# Study of Out-Of-Hospital Access to HIS System: A Security Perspective

**DOI:** 10.3390/s19112628

**Published:** 2019-06-10

**Authors:** Chih-Yung Chen, Yi-Chang Hsu, Chang-Ching Lin, Jeyhun Hajiyev, Chia-Rong Su, Ching-His Tseng

**Affiliations:** 1Department of Multimedia Design, St. John’s University, New Taipei City 251, Taiwan; yung@mail.sju.edu.tw (C.-Y.C.); m9244107@gmail.com (Y.-C.H.); 2Department of Information Management, National Defense University, Taipei City 112, Taiwan; 3Department of Management Sciences, Tamkang University, New Taipei City 251, Taiwan; tclim8@mail.tku.edu.tw; 4Department of Information Management, Chang Gung University, Taoyuan City 333, Taiwan; ceyhunhajiyev@gmail.com (J.H.); thomastseng1970@gmail.com (C.-H.T.); 5Department of Information Technology, Takming University of Science and Technology, Taipei City 114, Taiwan

**Keywords:** m-health, user authentication, SMS OTP, dynamic account, privacy protection, encryption card

## Abstract

In light of the need for Extramural Hospital Information System (HIS) access through mobile devices outside the hospital, this research analyzes situational information security threats, including the circumstances in which a mobile device may get lost and personal data may be stolen. Moreover, the system needs to be implemented in accordance with the regulations. Based on the security threat analysis, it is proposed to use a security control module to provide a security-enabled HIS proxy module, two-way authentication module, and One-Time Password (OTP). The sending module and cryptographic technology computing module with Micro SD encryption card form a set of HIS extension system, which includes the SMS OTP method to simultaneously verify the two-way authentication mechanism of a user and the device that the user owns.

## 1. Introduction

Since the implementation of the National Health Insurance (NHI) in Taiwan in 1995, the majority of medical institutions have established a fairly comprehensive HIS. Mobile health (m-Health) powered by smartphones and tablets highly improves the efficiency of medical care and protects health care staff who have relatively limited access to desktop computers. It also allows medical staff to access to medical treatments and HIS system for observation-related procedures.

Kristen [1] reviewed data security and privacy policies for the mobile application (App) designed for the monitoring of depression. Most Apps reviewed by Kristen are not transparent enough regarding the data security information. Laura [2] proposed a guideline for smartphone usage in the health care setting. The health care providers must be trained on how to reduce distractions caused by personal devices and to minimize them in a clinical setting. Chester [3] referred m-Health as the utilization of mobile devices (e.g., cellphones and tablets) for the delivery of health care. The author also stated that 83.3% of the respondents believe that m-Health presents diverse opportunities to improve health care delivery. In addition, 95% of the respondents indicates the potential for its future usage. Shaidah [4] proposed the innovative approaches in mobile setting for addressing complex health concerns, which are intended to be used outside clinics. Current m-Health Apps are still vulnerable regarding security, usability, and evaluation issues. Jonathan [5] recognized that mobile technology can improve the health of world populations. This technology has already improved the doctor-to-patient interaction by reducing costs and improving care for patients. Fatima [6] recommended that m-health has successfully replaced the conventional delivery system of health care, allowing continuous and pervasive health care at anywhere and anytime. Chronic disease Apps are increasing, as many health workers, patients, and clinicians already embrace smartphones in their comprehensive and diverse practices. One of the m-health’s biggest fears and technical barriers is the security and privacy of the personal information. In-depth analysis shows that the most significant challenge confronting is the Bring Your Own Devices(BYOD) policy, and very little has been done to tackle with this security challenge [7]. Daniel [8] proposed mobile Apps used in medicine with the aim of providing a personalized solution to disease management. Li et al. [9] proposed SEMAAC, which is a secure and efficient multiauthority access control system for IoT-enabled mHealth. It reduces the user decryption overhead, while the decryption work is executed in cloud server. Xiong et al. [10] suggested privacy and availability data clustering (PADC) scheme based on k-means algorithm and differential privacy. The scheme satisfies the goal of differential privacy and prevents privacy information disclosure.

Nowadays, most m-Health systems are designed in accordance with the framework of a hospital intranet or interagency VPN extranet, and in many cases where network data access services are advertised to be ubiquitous, the hospital improves the server. Besides the integration of medical information and adoption of appropriate information security measures [11], it is necessary to provide more convenient out-of-hospital HIS access service for medical personnel. By doing so, operators can implement medical services more effectively and efficiently.

However, while considering the implementation of HIS access service, as medical information is related to personal privacy and ultimately becomes smart information, the hospital’s security requirements for medical information access are no less than other smart institutions [12]. In order to access medical information from the Internet Mobile Network (PLMN) or other ISP networks, more stringent protection measures are mandatory, which can also prevent unauthorized access to sensitive information due to the extent that mobile device identification code, IP address, and other digital information may be mistreated, while at the same time the user’s password may easily be stolen. It is a major concern that must be dealt with prior to introducing an out-of-hospital HIS access system.

The Taiwan Personal Data Protection Act was officially launched on October 1, 2012, and medical resources were broadly included in the list of restrictions. Hence, in response with the litigation of the personal capital incident, medical institutions bear the burden of proof. The responsibility for the protection of resources is a function that must be included when the hospital provides an out-of-hospital HIS access system. It is also a major consideration for the system to be put into an actual implementation. Henceforth, information security protection measures are only used in Taiwan to access the structure of medical information systems on the intranet. In order to improve the quality and timeliness of its medical service and to extend out-of-hospital mobile devices to obtain medical information, it is vital to analyze information security threats and weaknesses in the postopening usage situation and select the most appropriate information technology tool to enhance the scope of protection.

The main contribution of this research is to provide physicians with portable mobile devices for effective and quick diagnosis and problem solution through the network connection, and in case of limited access to resources. By considering that the transmission of patient information and private data may undergo security threats, it is necessary to take security protection measurements with the implementation of new technology. Hence, this study proposes the OTP concept, so that medical personnel can securely access patient data outside the hospital.

## 2. Demand Analysis

### 2.1. Information Security Threat Analysis of HIS Access

The out-of-hospital HIS access system can be regarded as HIS extension subsystem that can be paralleled with the hospital’s original HIS system. Security threats to the system and its location must be initially explored for an improvement of the system. The source of the security threat must be analyzed from the perspectives of user, Internet, as well as a system. Prior to service provision through information system, the authentication of login system is executed to verify whether or not a user can provide the private information with the authorization. Therefore, the authentication system of information includes the information that is stored on a user device, and must be properly designed so that malicious third parties cannot impersonate the real user to log in to the system. The threats for user, Internet, and server are classified in Figure 1 below.

The main threats on the user-side include sneaking into the password, placing a Trojan horse virus to copy the password, and a malicious third-party who steals the file (including the information temporarily stored by the browser) [13]. Internet-side threats include malicious third-parties who use the information recorded on the user-side to impersonate the real user to log in to the system and copy the password [14]. Finally, the system-side threat refers to the malicious third-party intrusion in to the database [15,16] or the inappropriate behavior of internal personnel who may expose the information [17]. Moreover, the attacker may misuse authentication process to block the entire service.

Although information security threats are externally possible attacks, they are not equal to information security risks. By improving the weaknesses in the use chain, security risks can be effectively reduced. Therefore, the out-of-hospital HIS access system must be designed to reduce the weaknesses of the technical security solutions described in the previous section and allow management units to improve overall security protection capabilities.

The risk of the leakage of internal employee data is higher than that of external data. Along with the above-mentioned protection of security threats of the HIS access subsystem, technical solutions of the HIS access subsystem must be considered. Although the mobile device of a user is stolen and malicious third parties diffuse into the system, the medical system can still protect actual user data.

### 2.2. Information Security Technology Mechanism and Its Analysis

Since the development of information security technology mechanisms, the common principle of design is based on cryptography: (1) hash operation, (2) symmetric cryptography, and (3) asymmetric cryptography. Secure Hash Algorithm (SHA) is characterized by accommodating inputs of any length. The output is called a hash value and is of fixed length. The current SHA-256 is used by banking industry. For instance, the length is fixed to 256 bits, which is equal to 64 characters. The hash operation has the collision resistance property, and it is computationally hard to reverse the input value from the hash value. Therefore, the hash value is also regarded as a representative of its input value, so-called “digital digest” or “digital fingerprint”. In the application of user authentication, the hash value of the user password can be used as a substitute value to strengthen the weakness of the password.

Symmetric cryptography is a method for encrypting and decrypting information. It is characterized by the similarity between secret key of encryption and decryption. Currently, Advanced Encryption Standard (AES) is widely used, and the secret key length is 256 bits [18,19]. As AES adopts “replacement–combination architecture”, it has good execution efficiency on both software and hardware. In the application of user authentication, the device at the user-side uses the password hash value as an encryption key to encrypt a random number given by the system-side, while the encrypted result is transmitted to the system. The system verification method uses hash code of the password registered by a user as decryption key to decrypt the received information or data. If the outcome of the decryption matches the random number, it is inferred that the user provides the correct password, so the user is allowed to log in the system. It is called “challenge and response” mechanism.

The RSA method [20] is a mainstream technology of asymmetric cryptography. It has two main purposes: (1) information protection and (2) electronic signature production and verification. The RSA method uses two keys, namely public and secret key pairs. The public key can be publicly available without having to maintain confidentiality, whereas the secret key is protected by a user. Messages that are protected with public key encryption can only be decrypted using the corresponding secret key. Another function of the secret key is to make an electronic signature, where the validity of the signature can be verified by using the corresponding public key. The RSA electronic signature is the basis of medical signature, mainly as a proof that the digital information is generated or altered. For instance, a physician signs the medical certificate by using a medical signature. RSA encryption and decryption are not used to protect lengthy information. It is because RSA’s modular exponential calculation cost is relatively high with low computational efficiency. However, the application is used to securely transmit the AES key. Encryption and decryption use the AES method, corresponding to the basic principle of SSL operation.

Biometric methods, such as fingerprints, face recognition, and authentication irises, are applied to the system protection against malicious third-parties. Another technology that is considered safe is OTP. It is different than regular password. The password is usually determined by a user, and has a meaning, which is not limited to a single usage. The single password was first seen in the study of Leslie Lamport in 1981 [21]. It was emphasized that if the password is limited to a single use, it must be inferred the next time and after the user logs in. The method proposed by Lamport allows users to retrieve a series of OTPs by memorizing a password. The computer system only needs to record a verification value. The core of this method is to use the hash function to calculate a series of orders, such as the password and corresponding system verification value. However, each time the user logs in, multiple hash calculations must be performed continuously, indicating that the application has an uncertainty regarding the efficiency. After a certain number of entries, there are restrictions on it. In practice, there are two commonly used OTP mechanisms: (1) A user reads an OTP generated by a “Password Generation Token” and inputs the OTP in to the connected device, which is transmitted to the system-side as the basis for transaction confirmation. (2) Short Message Service (SMS) OTP: The system uses the SMS to transmit OTP to the user’s mobile device and instructs the user to enter the OTP in a specified time. The user’s connected device is transmitted back to the system for verification. The token user in the first method has relatively high security, but the disadvantage is high-cost, while the user must carry the token at any time for using in different places. The second way is that the system uses the short message to transmit the OTP to the user, which has high acceptability rate and is less costly. Because, mobile device is a widespread handheld device, and the infrastructure of the mobile communication service is well-established. Receiving a short message is a basic function of a smartphone, and the transmission channel of the short message is a transmission path outside the Internet, which can reduce the risk in the system.

## 3. Information Security Technology

### 3.1. SMS OTP Mechanism

In the short message OTP user authentication mechanism, when the system receives a service request from a user through the user device, the system generates a random number as the OTP for login verification. Then, the system uses the short message service to transmit the OTP for the use. On the mobile device, the user is required to input in the user terminal within a certain time limit, which is transmitted back to the system for verification. If the system can receive a matching OTP within a certain time limit, it is allowed to log in to the system. The time limit for entering the OTP must conform to the convenience of the user and security of the system. Take the online banking of Standard Chartered Bank as an example [18], the time limit is set to five minutes, whereas some other banks set a relatively longer time interval, giving users greater convenience. Figure 2 represents the implementation process of the SMS OTP user authentication mechanism. The dotted arrow in the figure acts for “message path” used by the SMS service. The message path is the communication path used by the mobile device, which is different from the network path used in the information system service process. Even if the current trend is that the mobile and computer communication are integrated and mobile device acts as the user terminal that can access the Internet. The transmission path of the message transmission channel and the information system message of the device are still not identical.

The user makes a request to use the service; this is the first step in Figure 1. The request is to use the device to communicate with the system. The most common client device is a personal computer operated by the user or smartphone and tablet used to receive the newsletter or notification. When the system receives the user’s login request, the system generates a random number as the OTP that the user must use to log in, and then transmits the OTP to the user’s mobile phone by means of a short message service. Here, the system-side must know the number of the mobile phone. This information is entered to the system by the user.

As shown in the flow of Figure 2, after transmitting the OTP newsletter, the system immediately instructs the user to read the short message received by the mobile phone, and inputs it to the user device by the user terminal to transmit back to the system, indicating that the message is received. The transmission path is an Internet communication path established by the user device and the system, while the path of the transmission of the short message is different.

The user operates according to the instructions of reading the short message to obtain OTP, inputting the OTP to the user terminal device, and transmitting the input back to the system by the user terminal device. When the system receives the OTP returned by the user device, the system compares the received OTP with generated OTP, and allows the login when the matching result is obtained. If the system receives the OTP after the time termination, or if the received OTP does not match the generated OTP, the user is usually instructed to perform login again, while if the number of failed login attempts exceeds the upper limit, user is refused to log in. According to the transmission protocol of the SMS service, the content of the SMS is encrypted and protected during transmission. This feature enhances security. More importantly, the main advantage of the SMS OTP user authentication mechanism in security is that the OTP is limited once and has a limitation on the usage time, which can effectively defend against the guessing attack on the user-side.

The short message OTP mechanism has advantage and is one of the mainstream methods, while there is a room for improvement in terms of user convenience or security. The user may refuse to enter the mobile phone newsletter or accidentally touch other buttons so that the OTP input is rejected by the system. In this case, the system usually asks for the entry again. If the number of failures reaches the predefined limit, the program that enters the OTP will be rejected. In more rigorous systems, such as online banking systems, their security regulations generally have an upper limit on the number of failures to confirm OTP. If the upper limit is exceeded, the system-side will further suspend the permission to user account. The user must carry the identity document to the service provider’s service base to unlock the account. It is worth noting that if the upper limit of the number of failures is set to 1, the system will suspend the user account. It is an unreasonable psychological burden for the user, which also increases the customer service cost.

In the next scenario, the upper limit of the number of failures is greater than 1. There can be two possibilities: (1) When the system verifies that the OTP is wrong or the OTP is invalid because of the timeout, it will send a new OTP message to the user’s mobile phone, asking the user to use the new OTP to perform login verification again. The risk of system intrusion may increase the inconvenience for users. (2) The system only sends message back, indicating that the OTP is entered incorrectly and asks the user to correct the input. This situation exempts the user from the opportunity to read the new OTP but gives the attacker a chance to guess the OTP. Hence, the attacker can assume that the user only enters one or two characters in the OTP incorrectly, and the login verification only fails once, and the attacker has an opportunity to conduct a guessing attack. At this point, guessing the probability of successful attack should be much easier than cracking the OTP generated by the system without information.

### 3.2. Strengthening Security Challenges and Response Mechanisms

The traditional “challenge and response” technique is commonly used in the user authentication mechanism. The main purpose is to prevent the user’s password from being transmitted on the network and being recorded. This mechanism uses symmetric and cryptographic encryption and decryption computations (such as the AES calculation), and its concept is shown in Figure 3. The “cryptographic calculation” is used in the “challenge and response” process, where the input includes the plaintext and the encryption key, and the plaintext encrypted output is called chipper text. To restore the chipper text back to the plaintext, the “decryption calculation formula” must be used, while the same encryption key must be used to encrypt the plaintext as the decryption key to restore the original plaintext.

In the “challenge and response” technique, the user uses the secret key corresponding to the system-side, where the result of the system-side decryption matches the challenge message. If there is a match, it can be inferred that the user-side uses the correct one. In terms of the secret key, the user and the system must share the secret key. In general, the secret key is the hash value of the user-defined passphrase, and this hash value is also registered and stored on the system-side.

In the information system, if the hash value stored on the system-side is obtained, the attacker can skip the step of calculating the hash value of the password on the user-side and use the obtained hash value to challenge and respond to the system-side, which elaborates security risk.

The challenge of enhancing security mechanism of response is shown in Figure 4. The basic assumption is that the user holds a secret key, and the system stores the hash value of the private data as the verification value. The core concept of verifying whether the user’s input on the user-side is consistent with the user’s secret key is that the result of decrypting the response message by the system-side must restore the user input. Therefore, the user input from the response message is decrypted by the system, and if the hash value matches the hash value stored on the system-side, the user input is correct. The “user secret key” here can be a passphrase or an OTP. In response to the calculation of the message, the user secret key must be used, but the system only stores the hash value of the user private data, and therefore the hash value of the user private data cannot calculate the response message. It is effective in improving the security of the traditional challenge and response mechanism.

The symbol ⊕ in Figure 4 represents the “XOR (Exclusive OR) calculation”. In Figure 4, the challenge message is a random number generated by the system. The response message is calculated by using the hash value input of the user with an encryption key, and the encrypted plaintext is “challenge message & user input”. The result of the calculation and the decryption response message of the system-side is the decryption key of the stored user secret key, and the decryption result is “recovered (challenge message & user input)”. By using the originally generated challenge message the system-side can calculate (challenge message ⊕ restored (challenge message ⊕ user input)), obtain the restored user input, and calculate its hash value. If the calculated hash value and decryption key (e.g., stored user private data hash value) are consistent, the user’s input is correct.

### 3.3. Method and System for Protecting Digital Secrets

The technique derived from the document “Partition and Recovery of a Verifiable Digital Secret” of the Republic of China Patent No. I255121 is described in detail in Figure 5. Its content contains three specific conversion functions, namely f_1_, f_2_, and f_3_.

By converting the functional formulas f_1_ and f_2_, a digit secret key can be divided into two digits, called “secret-dependent digital segment” and “secret-independent (secret-independent) digital segment”. When the digital secret key is to be entered, the above two-digit portions must be received, and then calculated by the conversion function formulas f1 and f3. By dividing the digital secret key, the protected digital secret key is stored in persistent memory. The holder of the secret data can easily enter the correct personalized secret and reply to the protected digital secret from the digital secret related part.

Verifiability is an important concept of “method for protecting digital secrets and its system”. For instance, when splitting a protected digital secret data, such as a symmetric cryptographic secret key, a one-way hash function can be used. Calculating the hash value of the digital secret data, using such a function cannot reverse the characteristics of the calculation, to ensure that the hash value of the digital secret cannot push back the corresponding original digital secret. Moreover, such a function has a crash impedance characteristic, and it is impossible or very unlikely to input two different digit secret data to get the same hash value. Therefore, the hash value of the digital secret data can be stored in advance as the verification information when the digital secret data is restored, and it is confirmed whether or not the correct digital secret data is restored.

If the digital private data is a private key of an asymmetric cryptography, such as the private key of the RSA cryptography, according to the characteristics of the RSA cryptography, the message encrypted by the public key can only be solved by the paired private key. Therefore, in response to the verification of a protected private key, a message “M” can be encrypted with the public key paired with the private key, and then decrypted with the private key. If the decrypted result matches the M, it is confirmed that the private key entered is correct.

### 3.4. User Authentication Method and Storage Medium by Linking Randomly Generated Authentication and Personalized Secret Data

The “User Authentication by Linking Randomly-Generated Authentication Secret with User-Chosen Secret Password” is a user authentication technique extending from “Methods and Systems for Protecting Digital Secrets” by the Republic of China Patent No. I293529. The content of the file includes the technology of “user-side verification program” and “user identity authentication program” performed by the client and the server system respectively, including a registration program, a login program, and a user-side verification program, which are described in the following.

There are two main tasks involved in the registration process. Initially, the user registers a hash value with the designated secret system as the registered secret. Then, the user prepares an authenticator for use by the login process.

The detailed process of the first task is shown in Figure 6 and the following steps.
(A)The user enters the user identification name and the system identification name on the user-side. The system identification name is mainly used to determine the server system to be registered. The user identification name, such as the account number, must be entirely or partially the identification information of a particular user, mainly as a basis for retrieval.(B)A digital random number is generated as the authentication secret on the same user-side, denoted by SA. In some embodiments, the authentication secret may also be generated by a server system.(C)The hash value of the authentication secret is calculated by the same user-side according to the selected one-way hash function and is represented by hash value (SA).(D)The user identification name and the authentication secret hash value Hash(SA) are transmitted to the server system corresponding to the system identification name at the same user-side.(E)The system-side stores the received secret hash value as a registration secret, together with the received user identification name in the authentication database.(F)The system sends a registration confirmation message to the user.(G)The final step is characterized by registering the program on the same user-side.

The second step is executed following the first step, and receives the authentication secret SA, the authentication secret hash value Hash(SA), the user identification name, and the system identification name. This step is mainly used for preparing identification data for the user and storing it in persistent memory. In order to achieve mobility and security, a personalized handheld device, such as a memory card or mobile phone, can also be used as a storage medium for storing authentication data.

The detailed process of the second step, as shown in Figure 7, is described as follows.
(A)After obtaining the authentication secret SA, the user identification name, and the system identification name, the user inputs the personalized secret SP, which is also referred to as the first user secret. The personalized secret SP and the authentication secret SA are independent of each other.(B)The user secret data is then calculated with the calculation formula SU = f_2_(f_1_(SP), SA), which is represented by SU. The client secret SU is also called the second user secret; f_1_ and f_2_ are two functions for sharing the secret data. For the nature and example formula, refer to the “verifiable digital secret segmentation and response” described above.(C)Integration of the client secret SU, the user identification name and the system identification name into an authentication data piece.(D)Identification of data that is stored in a storage medium(E)Calculation of the secondary hash value Hash^2^(SA) of the secret SA according to the selected one-way hash function.(F)Adding the secret secondary hash value Hash^2^(SA) to the authentication data.

Hash^2^(SA) is used instead of Hash(SA) as verification information to avoid duplication of information. According to the one-way hash function, it is computationally incapable of reversing. Leaking Hash^2^(SA) does not help guess Hash(SA), which is the secret data registered on the system.

After the user has registered with a specific system, the user’s login process is shown in Figure 8 and Figure 9. The details are further presented in the following.
(A)The user enters a personalized secret SP input and the system identification name.(B)Using the received system identification name as an index, the user terminal can retrieve the corresponding authentication data piece from the personal login file in the storage medium and select the user secret SU and the user identification name.(C)The user performs the calculation of f_3_(f_1_(SP), SU) to reply to the authentication secret. Here, f1 and f3 are two functional formulas for replying to the secret of identification. The properties and example formulas also refer to the content of the “verifiable digital secret segmentation and reply” described previously.(D)The user computes the user-side hash value from the replied authentication secret using the selected one-way hash function.(E)A second hash value is calculated from the user-side hash value, expressed as Hash^2^(SA).(F)The user retrieves the authentication secret secondary hash value Hash^2^(SA) from the authentication data piece according to the system identification name.(G)The user compares whether the calculated quadratic hash value Hash^2^(SA’) is equal to the retrieved secondary hash value Hash^2^(SA).(H)If the result of the comparison is consistent, the subsequent steps are continued; otherwise, the user is required to reimplement the login procedure.(I)The user makes an access request to the system to initiate the “challenge and response”. The system here is determined by the system identification name obtained in step (A).(J)After receiving the access request, the system that is determined to access generates a random message as a challenge to the user.(K)The system transmits a challenge message to the user.(L)The user uses the hash value generated in step (D) as the encryption key of the encryption challenge message to generate a response message.(M)The response message and the user identification name obtained by step (B) are transmitted by the user to the system.(N)The system receives the user identification name and response message.(O)The system uses the user identification name to retrieve the registration secret.(P)The system uses the retrieved registration secret as a decryption key and decrypts the response message to produce a decrypted result.(Q)The system compares the decryption result with the challenge message.(R)The system resolves to authorize or deny the access request and passes its decision to the user. When the comparison result matches, the access request is allowed; otherwise, the access request is denied.(S)The user receives a message from the server system for permission or rejection.

The process of the first and second changes of the secret data is described as follows (see Figure 10).
(A)The user logs in to the server system, whose registration secret is to be updated with the original first user secret (indicated by SP) and the second user secret (indicated by SU). The login process is the same as the login procedure described above.(B)The original authentication secret (indicated by SA) is changed to a new authentication secret (indicated by SA_new_). The new authentication secret may be a digital random number generated by the user, as previously described for the method of generating an authentication secret in the registration procedure.(C)Calculation of the new user-side hash value Hash (SA_new_), which is expected as a new registration secret.(D)The user uses the original user-side hash value, Hash (SA), as the encryption key to encrypt the new user-side hash value Hash (SA_new_).(E)The user transmits the new user-side hash value and the user identification name to the server system.(F)The server system obtains the encrypted new user hash value and user identification name.(G)The server system obtains the original registered secret hash (SA) by the user identification name.(H)The server system decrypts the encrypted new user hash value using the original registered secret as the decryption key.(I)The server system replaces the original registered secret with the new user hash value Hash (SA_new_) as a new registration secret.(J)The server system transmits a confirmation message to the user.(K)Following the confirmation message, the user divides the new authentication secret as follows; (1) in the first updated method, the new user secret is initially obtained, denoted by SP_new_, and then the following expression is calculated. The second user secret is updated, indicated by SU_new_, which refers to updating the client secret SU_new_ = f_2_ (f_1_ (SP_new_), SA_new_). (2) In the second update method, the first user secret is kept unchanged, the second user secret is updated by calculating the following expression: SU_new_ = f_2_ (f_1_ (SP), SA_new_).(L)The new user secret SU_new_ is stored in the storage medium by compiling the authentication data piece and replacing the original user secret SU.

When the user wishes to perform the third method of changing the secret, the user also needs to reply to the currently used authentication secret SA with the original first user secret SP and the second user secret SU. The user secret SP_new_ and the reply authentication secret SA are used as input values to obtain a new second user secret SU_new_, namely: SUnew = f_2_ (f_1_ (SP_new_), SA). If the user can perform the “user-side verification program” as shown in Figure 3, Figure 4, Figure 5, Figure 6 and Figure 7 to confirm the correctness of the secret, the change procedure can be performed separately on the user-side. Otherwise, the user must first log in to the server system and confirm the correctness of the identified secrets.

### 3.5. User Authentication System for Combining Single Passwords with Repeatable First Passphrases and Nonrepeating Second Passphrases

Document No. I374653 “The User Identification Technology and System for Combining a First Passphrase with a Repeatable First Passphrase and a Nonrepeating Second Passphrase” of the Republic of China, hereinafter referred to as the “Synchronous OTP two-factor authentication key”, the disclosure of which discloses a technique for performing OTP user authentication using a user’s passphrase, is based on the aforementioned patented methods of the Republic of China No. I255121 and No. I293529. The first passphrase is a password that the user decides and memorizes, while the second passphrase can be stored in a storage device, such as an USB flash drive and smart card.

The core of OTP technology is that every OTP is difficult to predict in advance. In general, using random numbers as OTP is one of the common methods. A random number with a sufficient bit length can be used as a strong passphrase, which is very difficult to guess, while another benefit is that there is no limit to the random numbers. Suppose P1, P2, … represent a series of OTPs used in sequence, V1, V2, … represent a series of computer system-side verification values corresponding to OTP in sequence, then each Pi is a random number, and each Vi is the corresponding Pi that is calculated by a hash function to obtain a hash value. If j is a positive integer not equal to k, then Pj and Pk will be different, and it is very difficult to derive Pk from Pj or Pj from Pk. According to this feature, it is very difficult to derive an unused OTP from a used OTP, which is one of the basic requirements of the OTP.

The content of the patented technology described in this section applies the content of the patented technology of No. I293529 of the Republic of China, including the registration procedure, the user-side verification procedure, the login verification procedure, and the password change procedure. Patent application No. I255121 is used for login verification. After the program, a “synchronous shift program” is executed to synchronously change the secret values of the user and the system, thereby generating the characteristics of the OTP mechanism.

When the user passes the login verification program, the system and the user perform a “synchronization process” to determine the secret key used to execute the challenge and response mechanism at the next login. The secret key to be used for the next login is determined by the user. It is usually a random number and will not be stored in the system. At the user-side, the new secret key and the first passphrase (e.g., the password remembered by the user) use the splitting expression to calculate the new second passphrase and update the old one, and then delete the secret key immediately. In the next login process, when the user inputs the first correct passphrase, reply operation can be performed along with the stored second passphrase to reply to the correct secret key. The hash value is on the system-side and stores the new secret key. In addition, the second hash value of the new secret key is the new user-side verification value. The execution process of the synchronous shift program is shown in Figure 11.

In Figure 11, the system sends a message to the user when the user successfully logs in. After receiving the message, the user generates a random number as a new secret key and calculates a new secret key, hash value, and second hash value. Next, the first passphrase and new secret key are used as inputs to execute the split expression, and the second passphrase is used for the next login, and then, the login is used. The hash value of the secret key is used as an encryption key to encrypt a hash value of the new secret key and transmit the encrypted result to the system. The way the system obtains a hash value of the new secret key is to use the stored verification value as a decryption key, decrypt the received encryption result, restore a hash value of the new secret key, and then restore the obtained hash value. The hash value is replaced with the verification value. After that, the second message is sent to the user to notify for updating the second passphrase.

After receiving the message of the second synchronization on the system-side, the user replaces the second passphrase for the login with the new second passphrase and updates the user authentication with the second hash value of the new secret key. After the user completes the update, sends a confirmation message to the system to notify the synchronization, meaning that the program is completely executed. Following the synchronous shift procedure, the user will have a new secret key. The changed key will not be stored on the user-side, and a new second passphrase will be stored. At the same time, a new verification value will be stored on the system-side. In other words, the synchronization pass program will be executed completely. Next, the second passphrase of the user and the verification value of the system will be replaced with new values.

A possible condition is that the system sends a second message to perform the synchronization change, whereas it does not receive the user confirmation message. Therefore, the user may have updated the second passphrase or the network may be unexpectedly interrupted. If the second message is received and the second passphrase is not updated, the technique of the patent publication utilizes an “alternate synchronization program” to overcome the above-mentioned situation in which the system-side may incorrectly reject the user’s login due to the fact that the synchronization shift program is not completely executed. It is described in a following paragraph.

Reviewing the aforementioned synchronization process, if the system sends a second message to perform the synchronization transition but does not receive the user confirmation message, the system-side notes “confirm synchronization tag” and retains the synchronization transition. The previous verification value is called the “alternate verification value”. The next time the same user logs in, if the secret key used by the system to perform the verification inference is incorrect but the confirmation of the synchronization is recorded, the system will use the “backup verification value” pair to verify. If the verification still results in an incorrect result, the user’s request is rejected. Otherwise, the synchronization change process is continued. After the synchronization change process is complete, the “confirm synchronization tag” is canceled.

The characteristics of the OTP user authentication technology described in this section are illustrated in Figure 12. The first passphrase of the user may be determined by the user’s autonomy or may be repeatedly used for multiple consecutive logins. Moreover, if the reading of the second passphrase is automated and does not require user intervention, it seems to the user that the traditional user login verification process has no operational burden.

### 3.6. User Authentication Method and System Combined with Dynamic Account

The operation of the information service system usually stores unique information in the system-side database to identify the user. The general name is the account number, identification code, identification information, and others. This section uses the “account number” for explanation.

In the user authentication process, the key to the user being authenticated with the system license is to be able to present the correct account number to inform the system of its identity. Moreover, the user must submit secret information corresponding to the account, such as a passphrase, OTP, or fingerprint to prove that the user is indeed the owner of the account. On the system-side, as large number of users must be served, the authentication information of individual users must be stored in the database. The representative of this verification information is generally the user account. Therefore, when the system receives the account proposed by the user, the account is indexed, and then the authentication information required by the user is searched.

In the user authentication method uses a passphrase and “challenge and response” mechanism, the account number is mostly transmitted to the system in plaintext format, and only the user’s passphrase is protected, such as using the OTP or “challenge and response” mechanism. In this design, if the Secure Socket Layer (SSL) connection is not established between the user and the system, the attacker may log the account and obtain the related information for login verification. However, the general user’s habit of using passphrases is that they are replaced after a certain period of time, and the passphrase is usually a short length of information. The attacker can try to log in to the system according to the account of the user-side record and guess the possible passwords. For such an attack mode, the common defense method is to limit the number of upper boundary of authentication failure. For instance, in most network ATM systems, when user input password fails for three consecutive times, the bank card will be locked for further use. The card holder must present the identity certificate to unlock the card. If the client and the system are operating in an SSL secure connection environment, the attacker can be prevented from logging to the account from the upper side of the network. However, the intrusion of the system-side database or the improper access of malicious attacker inside the system is also a possible way for the account to be obtained, and there is still a risk.

The technology of the “User Identification Method and System for Dynamic Accounts” described in this section has already been submitted for patent review of the Republic of China. The benefit is to avoid the above-mentioned account transfer in plaintext format and storage in the system-side database.

The technology of the “user authentication method and system combining dynamic account numbers” is designed by referring to the aforementioned three patent technologies. In this section, the user identification method using password and “challenge and response” mechanism is also used.

The current user authentication mechanism using a passphrase usually performs a “challenge and response” mechanism on the user-side and the system-side to achieve the purpose of not transmitting a passphrase on the network. At the same time, it is used to calculate the response message, that is, the chipper text obtained by encrypting the challenge message. The secret key used must be the hash value of the correct passphrase. On the system-side, the user’s passphrase is not stored; instead it is the hash value of the passphrase. Therefore, the system can find the corresponding password hash value according to the received account, and then use the found information as the decryption key to decrypt the challenge message from the received response message. According to the principle of symmetric cryptography, if the password entered by the user on the user terminal is correct, the challenge message obtained by the system-side decryption will be correct. On the other hand, if the password entered is wrong, the hash value of the password calculated by the user must not match the value stored in the system. The result obtained by the system from the response decryption will be inconsistent with the original challenge message.

The design described in the previous paragraph has security benefits. One is that passphrases are not transmitted over the network, and the second is that passphrases are not stored on the system. However, because of the execution of the “challenge and response” mechanism, the hash value of the passphrase is used as the secret key. Therefore, as long as the passphrase hash value is obtained, the user can skip the step of calculating the hash value of the passphrase and use the obtained hash value. Security risks can be effectively reduced by the “challenges and response” mechanism described above, while reducing the risk of password is required by guessing attacks after the account. The user can guess the passphrase according to the obtained account number and is more likely to guess the passphrase offline according to the obtained passphrase hash value.

By reviewing the user authentication mechanism using the passphrase, the information stored on the system-side contains at least two pieces of information: (1) the account number and (2) the passphrase hash value. Based on the core concepts of the patented “Methods and Systems for Protecting Digital Secrets” that have been described above, two pieces of information can be used to respond to a protected message. According to the patented method of “User Authentication Technology and System Using a Repeatable First Passphrase and a Nonrepeating Second Passphrase to Form a Single Passphrase”, the synchronization technology in the content can be used to allow the user to the system-side, synchronizing the information that is replied to the response, producing an OTP-like feature. Therefore, in accordance with the technical principle of user authentication of a dynamic account, it is additionally required to store the “dynamic account hash value” and a “password hash value” on the system-side server. The user maintains two secret values: (1) the “password” remembered by the user and the “account secret” stored in the storage medium. The account secret can be a random number. The synchronous change of the login verification and the stored secret value is as follows.

First, the user device makes a connection request to the system-side to receive the challenge message sent by the system. Next, the user is asked to enter a passphrase and calculate the passphrase value. The password is hashed to the secret key, and then the user device further calculates the response message and uses the password hash value as well as the account secret as the input of the reply calculation formula to reply to the “dynamic account” and calculate the hash value of the dynamic account. After that, the user device calculates the hash value of the “dynamic account hash value || challenge message”, and the result is called “obfuscated account”, and is sent to the system-side together with the response message. The “||” symbol here represents a concatenation operation.

After receiving the response message, the system calculates the hash value of the stored “dynamic account hash value || challenge message” until the result of the calculation can be consistent with the received obfuscated account. The decrypted key is retrieved by the hash code of the passphrase, and the system performs the decryption of the received response message. If the result of the decryption does not match the challenge message, the login is refused, otherwise the login is allowed and the synchronization process is continued.

In the beginning of synchronous shift, the system transmits a message to perform the synchronous transition to the user device. After receiving the message, the client generates a random number as a new dynamic account and calculates the hash value of the new dynamic account. After that, the pass-through passphrase hash value and the new dynamic account number are used as inputs to execute the split-calculation formula, and the account secret to be used for the next login is calculated. Then, the passphrase value is used for the login as an encryption key, encrypting the hash value of the new dynamic account and transmitting the encrypted result to the system.

The system obtains the new dynamic account hash value by using the stored password hash value as a decryption key, decrypting the received encryption result, restoring the new dynamic account hash value, and replacing the hash value obtained by the restoration. The original verification value is used. After that, the second message is sent to the user to notify for updating the account secret.

After receiving the message of the second synchronization on the system-side, the client secretly replaces the account secret used for the login with the new account. Once the update is completed, a confirmation message is sent to the system-side to inform the synchronization that the program has been completely executed.

Since the new account is determined by the user, in actual design, the decision of the new dynamic account and the transmission of the hash value can be performed simultaneously when transmitting the obfuscated account and the response message.

There is a possibility that the execution of the synchronous transition is performed: the system sends a second message to perform the synchronization transition but does not receive the user confirmation message. Therefore, the user terminal may have updated the account secret or may not receive the second message due to an unexpected network interruption, and thus does not update the account secret. Therefore, this technology also applies the “alternate synchronization program” that uses identification technology and system combining the first pass password with the repeat and the second pass password that is not repeated to form a single pass password. The program is not fully executed, which may cause the system to erroneously reject the user’s login.

The characteristics of the dynamic account authentication technology are described in Figure 13.

## 4. Construction Solution

### 4.1. System Architecture

The major goal regarding safety is that when the physician’s mobile device is stolen and the attacker has obtained the information used by the system to authenticate the login, there is still no immediate risk of information leakage. For detailed items, please refer to the security requirements of medical institutions that use mobile devices to access medical information outside the hospital.

The system architecture is shown in Figure 14. The authentication mechanism adopted is based on the authentication designed by the background management system, and the information security needs are added to form a two-way authentication that can simultaneously authenticate the user and mobile device. In addition, the Micro SD card can also be used to provide the security of each mobile phone in the voice telephone system.

In Figure 14, the system-side includes a security control module, a two-way authentication module, a cryptographic technology computing module, a Micro SD HSM card, and an OTP transmitting module. The user device includes a mobile phone, a tablet, and a Micro SD encryption card. The process of identifying the user and device as well as the functions of the individual modules and devices is described below.

### 4.2. The Process of Identifying Users and Their Devices at the Same Time

On the system-side, the security control module is responsible for receiving the connection requirements of the mobile device. If the device connection uses the in-hospital network that has been verified, the security control module skips the authentication mechanism required for out-of-hospital connection. Hence, the agent medical system provides the service, otherwise it is sent to the two-way authentication module to start the authentication process of the user and device. The OTP generated by the authentication process is utilized by the cryptographic technology computing module called Micro SD HSM card, through OTP. The sending module communicates to the designated platform and transmits the OTP to the designated mobile phone by means of a short message. At the same time, the cryptographic technical computing module requests the Micro SD HSM card to perform the encryption and decryption used by the authenticating user and device. The connection information that has not been authenticated will be blacklisted, and the authenticated mobile device will receive information identifying the correctness of the system, allowing the user know that the system is connected. At this time, the information used by the system to authenticate the user and its device is automatically updated. Attempts to use old data to impersonate the user to log into the system will not succeed.

The process at the user’s end: First, the user executes the connection App on the mobile device and enters the passphrase as indicated to suggest the need to use the medical system outside the hospital. When the connection App receives the passphrase, it will send the passphrase to the Micro SD encryption card, request the original information of the memory used by the encryption card, and calculate the account used for verification to the system. When the result is calculated, the encryption card obtains the dynamic account verification value of the memory and checks the correctness of the calculation result. If the result does not match, the user is required to re-enter the passphrase. Otherwise, the calculated account number is transferred to the security control module by the connection App, and the system is required to transmit the SMS OTP to the user’s mobile phone and request the system-side. Two pieces of information are returned: (1) to preverify whether the user has entered the correct OTP and (2) to provide the user with a visual check to see if the correct system is connected. The message to check the correctness of the system is encrypted, and the decryption key is only restored when the user enters the correct OTP.

In the next step, the connection App sends the received preverification information, the system correctness message chipper text, the user-entered OTP, and the previously verified passphrase to the encryption card, while the encryption card calculates the secret value required for the login system. When the calculation result is checked by the preverification information, the calculation result is used as the decryption key, and the chipper text is decrypted from the system correctness message into the original plaintext message. Finally, the login secret value and the system correctness message are sent back to the connection App.

Before the connection App sends the login secret value to the system-side for verification, the system correctness message will be displayed to the user for visual inspection. At this point, the user can suspend the execution of the authentication program or allow the connection App to continue execution. After the connection App sends the login secret value, if it can receive the message allowing login, the user and the mobile device will be verified, while, at the same time, the original information for calculating the account and the login secret will be updated, and then the confidential application and the specified browsing will be opened. Browsers, such as Chrome and Mozilla Firefox, can be used. The browser is used to obtain the medical service represented by the security control module, and the confidential application monitors the execution of the browser. When the browser is closed, the medical information that is temporarily stored in the mobile device is cleared to prevent others from stealing the data.

### 4.3. Description of System-Side Function Module

The functions of the module executed on the system-side are also illustrated in Figure 14 and Table 1, including a security control module, a two-way authentication module, a cryptographic technical operation module, a Micro SD HSM card, and an OTP transmission module. The security control module is a system that is open to the outside world. The main function is to manage the connection between the individual user device and the system end and forward the message to the relevant system server on the system-side for the relevant program to be executed. After that, it receives a reply message from the system server. The specific operation mode is explained below.
If the user receives a connection request from a tablet device using the in-hospital network, it will be sent directly to the medical system without changing the structure of the original intranet using the medical system.If the connection request from the device from the external network is received, the relevant information (e.g., the IP address) of the connection is forwarded to the two-way authentication module to activate the authentication process of the user and device.If the connection of the user is determined to be the authentication phase, the login information of the user device is forwarded to the two-way authentication module to verify whether the user and device are authorized.If the connection of the user is determined to have passed the authentication, the message is forwarded to the Web medical system.

The service is provided by the security control module agent medical system, which aims to hide the medical system from the outside and reduce the possibility of malicious attacks on the medical system. Considering the digitized information such as the mobile device identification code and the mobile device IP address, it may be altered to impersonate the authorized device. Therefore, the security control module identifies the connection between the internal network and the external network. The two network card entities isolate the internal and external network architecture and determine the path of subsequent forwarding information according to the connected network card.

When the security control module agent medical system provides services to external network users, it is recommended to operate in the TLS/SSL environment. There are many ways to establish TLS/SSL [22]. For instance, Microsoft’s Internet Information Services (IIS) service, which has a built-in TLS/SSL service, can be checked and enabled by the system administrator. It is generally considered to be sufficient for the simultaneous connection of 200 people. The TLS/SSL environment can also be implemented by purchasing hardware devices such as hardware security modules (HSM), SSL acceleration cards, and SSL VPN devices. These hardware devices are available on the market.

The two-way authentication module is primarily responsible for identifying whether the user and its device are authorized. If the users and their devices fail to pass the authentication, they are blacklisted for control and refused to provide subsequent service provision. Users must file an application for blacklisting in accordance with administrative specifications. On the contrary, through the authentication connection, the authentication module will respond to the message allowing the connection, and return the user-registered security stamp, which is transmitted back to the user device by the security control module.

The program executed by the authentication module, if required to perform a cryptographic algorithm, is executed by the cryptographic technology computing module. In order to achieve the two-way authentication function, the authentication module must manage the information. The information is stored in the database in chipper text format, and the decryption key can be replied when the user is authenticated, but not stored in the system.

The cryptographic technology computing module is responsible for assisting the two-way authentication module to perform the specified cryptographic algorithm, including AES 256, SHA 256 [23,24], generating random numbers, and so on. The cryptographic algorithm calculation program executed in this system must be authenticated. The implementation of the cryptographic technology computing module has a variety of choices, depending on the number of users being authenticated at the same time, including the use of software programs to perform, or the use of hardware devices for assistance. Since the authentication process takes only a short period of time, the number of users being authenticated is less than the number of users using the medical system at the same time. This plan recommends the use of the MicroSD HSM card, which has been successfully used by national units.

The OTP sending module is responsible for managing the address of the SMS sending platform. At the same time, it is responsible for transmitting the mobile phone number specified by the two-way authentication module and the OTP to the SMS platform to transmit the OTP to the user’s mobile phone.

In addition, in order to manage the overall operation of the system, it is necessary to build a management module. This module contains administrative functions such as user, blacklist, Micro SD encryption card, and external network connection. At the same time, it provides a record of the connection system between the internal and external network.

### 4.4. User Device Description

The user devices include tablet devices, mobile phones, and Micro SD encryption cards.

The connection medical system application must be downloaded to the tablet device in advance. It is called “Connection App”. The App stores the external website of the security control module to reduce the possibility of users entering the wrong website, thereby reducing the chance of phishing attacks. At the same time, the tablet device must predownload a confidential App when it is authenticated through the system. After that, the connection App opens the browser and obtains the medical service.

The purpose of the confidential App is to immediately clear the information left by the medical system on the tablet device, and prevent others from stealing confidential information. When the confidential App is executed, when the user closes the browser or directly closes the tablet, the confidential App will execute the process of clearing the temporary data. Currently, some browsers (e.g., Mozilla Firefox), which can be executed on mobile devices, provide the user preferences setting option for automatically clearing temporary data when it is closed. Some browsers (e.g., Google Chrome) only provide the ability for users to manually delete temporary data. Different types of browsers provide different privacy protection methods, and the storage path of temporary data is also different. In particular, in the development of the Android operating system (OS), the OS before and after the upgrade usually has a big difference. Such phenomena must be ruled out to maintain the operation of the connection App and the confidential App.

This research combines the existing Mobile Device Management (MDM) tool in the hospital to limit the time for the medical tablet to upgrade the OS. It must be initially confirmed that the connection App and the confidential App can run on the new OS, and then open the medical tablet to upgrade the OS. The control of the OS version upgrade can also prevent the Micro SD encryption card from functioning properly. Medical personnel must be allowed to use medical services without interruptions or warning.

The Micro SD encryption card is a device for the use on tablet devices. With the Micro SD encryption card, the tablet device performs the computational burden of the cryptographic method and transfers to the Micro SD encryption card. At the same time, the encryption card has anti-hardware destruction capability, and can provide a secure storage area for storing confidential information. The user’s mobile phone is used to receive the OTP newsletter for authentication. The user’s mobile number must be registered prior to two-way authentication module. With the two-way authentication mechanism of the short message OTP feature proposed in this proposal, the user, the tablet device, and the mobile phone can be simultaneously authenticated, and all can meet the requirements of the medical system.

### 4.5. Operating Steps of Prototype System

The current study uses the use case description shown in Figure 15. The user uses the mobile device to access the HIS system through the security verification. The detailed system operation describing the execution of each module is explained in the next diagrams (see Figure 16, Figure 17, Figure 18, Figure 19, Figure 20, Figure 21 and Figure 22).

The encryption and decryption calculations used in the system operation are all AES 256. The hash value is the output value calculated by SHA-256. The symbols used in the description || represent the concatenation operation; ⊕ represents the XOR operation; hash represents the SHA-256 calculation; AES represents the AES 256 calculation; and the used key is indicated by the subscript. In addition, when describing the parameters required for the i-th identification process, the following i is indicated on the right-side of the parameter, and the subscript i+1 represents the next authentication process that has not yet been performed.
(A)Mobile number: The mobile phone number is registered by the user on the system-side and is used to receive the SMS OTP in the authentication program.(B)Device ID: The Device ID is the identification information of the tablet device, which is usually fixed when the device is shipped from the factory.(C)Pwd: Pwd is the password that the user chooses and remembers. The user can change it over time.(D)OTP: The OTP is generated by the system when the authentication program is started and transmitted to the user through the mobile phone newsletter.(E)Challenge message: The challenge message is a random number, generated by the Micro SD encryption card during each authentication process.(F)Dynamic account: The initial dynamic account is a random number generated by the system and is information that is not stored. In each authentication process, the dynamic account is the user’s identification information. According to the dynamic account, the department can find the relevant data of the corresponding user. The dynamic account is the primary key of the user data. The dynamic account number is not stored. It is replied through the calculation in the authentication process by:Dynamic account i = AES hash(device ID || Pwd)(Dynamic secret connection i)

In the different authentication process of the same user, the dynamic account number is not same, and it has the property of being used only once. Dynamic accounts are redetermined each time they pass system-side authentication by


*Dynamic account i+1 = AES Dynamic account i(hash(hash(Dynamic account i) || OTP i))*
(G)Dynamic account hash value: The dynamic account hash value is the information stored in the system-side authentication database. It is a hash value of the dynamic account, which is expressed as a hash (Dynamic account). The dynamic account hash value is updated with the decision of the new dynamic account each time the authentication process successfully verifies the user’s authorization. Considering that the connection between the tablet device and the system may be unexpectedly interrupted, in order to perform the authentication correctly, the dynamic account hash value before the update is temporarily stored, which is called the previous dynamic account hash value. When the system receives the mobile device, the message of the parameter is updated, and the previous dynamic account hash value can be deleted.(H)Dynamic account secret: The dynamic connection secret is the information stored on the Micro SD encryption card, which is updated as the new dynamic account is determined. The update method is
*Dynamic secret connection i+1 = AES hash(hash(device ID) || hash(Pwd)) (Dynamic account i+1)*
(I)Dynamic account self-validation value: The dynamic account self-validation value is the information stored in the Micro SD encryption card, which is the secondary hash value of the dynamic account, expressed as hash2 (Dynamic account). The dynamic account self-verification value is updated with the new dynamic account each time the authentication process successfully verifies the user’s authorization.(J)Confused after dynamic account pair: After the confusion, the dynamic account pair contains two pieces of information: the first value of the dynamic account after confusion and the second value of the dynamic account after confusion, which are temporary values calculated during the authentication process. The account is transferred to the system to prevent the dynamic account from being recorded. The confusing calculation method is
*First value = hash(hash(Dynamic account) || Challenge message))*
*Second value = AES hash(Dynamic account)(Dynamic account* ⊕ *Challenge message)*


In addition, during the system-side authentication verification process, the second verification value of the dynamic account after confusion is calculated. Hash value of the second value of the dynamic account after confusion, is expressed as


*Hash (the second value of the dynamic account after confusion)*
(K)Dynamic login secret: The initial dynamic login secret is a random number generated by the system and is information that is not stored. In each authentication process, whether the connection APP can correctly reply to the dynamic login secret is the technical key for the system to identify the user and its device. The formula for the connection APP to reply to the calculation of the dynamic login secret is as follows
*Dynamic Login Secret = AES hash(hash(deviceID) || hash(Pwd)) (Secrets of dynamic identification)*



In the different authentication process of the same user, the dynamic login secret is not identical and has the property of being used only once. Dynamic login secrets are redetermined each time through system-side authentication by


*Dynamic login secret i +1 = AES Dynamic secret login i(hash(hash(Dynamic secret login i) || OTP i))*
(L)Dynamic login secret hash value: The dynamic login secret hash value is the information stored in the system-side authentication database. It is a hash value of the dynamic login secret, which is expressed as hash (Dynamic secret login). The dynamic login secret hash value is updated with the decision of the new dynamic login secret each time the authentication procedure successfully verifies the user’s authorization. Considering that the connection between the tablet device and the system may be unexpectedly interrupted, in order to perform the authentication correctly, the dynamic login secret hash value will be temporarily reserved before the update, which is called the previous dynamic login secret hash value. When the system receives the authentication, mobile device updates the parameter information, while at the same time the previous dynamic login secret hash value can be deleted.(M)Secrets of dynamic identification: The dynamic authentication secret is information stored in the Micro SD secret card, which is updated as the new dynamic login secret is determined. It is expressed as
*Secrets of dynamic identification i+1 = AES hash(hash(deviceID) || hash(Pwd)) (Dynamic secret login i+1)*
(N)Device–user–OTP login code: The device–user–OTP login code is simply referred to as the OTP login code, which is a temporary value that can be obtained by both the system and the connection App. It is calculated as follows
*OTP login code = AES hash(Dynamic secret login || OTP) (Dynamic login secret ⊕ Challenge message)*



In the authentication process, the system-side performs the authorization verification of the user and its device according to the OTP login code calculated by the connection App.
(O)Device–user–OTP Self-verification code: Device–person–OTP self-certification code is abbreviated as OTP self-certification code, which is the temporary value calculated by the system. It is sent to the connection App to check whether the user inputs OTP correctly. The calculation formula isOTP Self-verification code = hash^2^(device–user–OTP login code)(P)System identification plaintext: The system identification plaintext is selected by the user and then stored in the system-side authentication database in ciphertext mode. The purpose of the system to identify plaintext is to allow users to visually judge the success of connecting to the correct system and reducing the probability of phishing attacks.(Q)System identification ciphertext: The system identification ciphertext is the information stored in the system-side authentication database and is also the type in which the system recognizes that the plaintext is encrypted. The encryption key is a hash value of the dynamic account, such as hash (Dynamic account). The system identification ciphertext is updated as the new dynamic account is determined. Considering that the connection between the tablet device and the system may be unexpectedly interrupted, the system identification ciphertext before the update is temporarily reserved, which is called the previous system identification ciphertext. When the system receives the message that the mobile device has updated the parameters, the previous system recognizes the ciphertext.(R)System identification restore ciphertext: The system identification restore ciphertext is a temporary value calculated by the system and is not stored. The calculation formula isSystem identification restore ciphertext = AES hash(Dynamic secret login) || OTP) (System identification plaintext)

The system identification and restoration ciphertext is transmitted from the system-side to the connection App for decryption, so that the system recognizes the plaintext for the user to visually judge the correctness of the connected system.

The above-mentioned parameters stored on the server-side include the following.
Mobile numberDynamic account with the previous hash valueDynamic secret hash value signed with the previousDystem identification with the previous ciphertext

The parameters stored on the Micro SD encryption card include the following.
Secrets of dynamic connectionSelf-verification value of Dynamic accountSecrets of dynamic identification

Parameters transmitted on the network include the following.
Challenge messageConfused dynamic account pairDevice–user–OTP login codeDevice–user–OTP self-verification codeSystem identification restore ciphertext

The operating steps of the prototype system are explained in the following.
User-to-server connection request
1.Ask [Connect App]2.Request enter [Pwd]3.Enter [Pwd]4.Response receive [Pwd]5.Request to return the medical system [URL], [Dynamic Connection Secret], and [Dynamic Account Self-Authentication Value]6.Return [URL], [Dynamic Connection Secret], and [Dynamic Account Self-Authentication Value]7.Receive [URL], [Dynamic Connection Secret], and [Dynamic Account Self-Authentication Value]8.Transfer [Pwd], [Device ID], and [Dynamic Connection Secret], request to calculate [Dynamic Account] and its secondary hash value9.Receive information and calculate [dynamic account] and its secondary hash value return10.Receive [Dynamic Account] and its secondary hash value11.Press [Dynamic Account Self-Authentication Value] to match [Dynamic Account Sekundär Hash Value].12.Is it typed over three times?13.N: Re-request [Pwd] (Go to step 2)14.Y: Stop using the appropriate message15.OK: Ready to connect16.Request to return a random number17.Pass back a random number18.Receive random numbers as [challenge messages]19.Transfer [Dynamic Account] and [Challenge Message] and ask to calculate [Dynamic Dynamic Account Pair after Confusion]20.Receive the information and calculate the [confusing dynamic account pair]21.Receive [confusing dynamic account pair]OTP generation by server after nonblacklist verification
22.Connect to the medical system and send [confusing account pair] and [challenge message]23.Receive [URL], [Confused account pair], and [challenge message]24.Dispatch routing: judged to be a new connection25.Transfer identity verification request, [confusing account pair], and [challenge message]26.Receiving identity verification request, [confusing account pair], and [challenge message]27.Obtain [this dynamic account hash value] by using the received [confusing dynamic account pair first value]28.[The dynamic account hash value]29.Transfer [Dynamic Account] and [Challenge Message], request to calculate [Second Verification Dynamic Account Second Verification Value]30.Receive the information to calculate [the second verification value of the dynamic account after confusion]31.Receive [Second Verification Dynamic Account Second Verification Value]32.Does [the second value of the dynamic account after confusion] and [the second verification value of the dynamic account after confusion] match?33.Check: Is it to check the synchronization annotation (Y/N)?34.Synchronization Mark (Y/N)35.N: Blacklisted36.Y: Go to 22 after receiving [last dynamic account hash value]37.[Last Dynamic Account Hash Value]38.OK: Temporary variables are used to distinguish between the ne wand previous parameters.39.Take the corresponding [Mobile number] according to the [Dynamic Account Hash Value] obtained.40.[Mobile number]41.Request [OTP]42.Generate six code random numbers as [OTP]43.Send [OTP]Sending OTP and preparing user verification test information by server
44.Receive [OTP]45.Send [Mobile number] and [OTP] to SMS platform46.Receive [Mobile number] and [OTP]47.Click [Mobile number] send [OTP]48.Send Complete message49.Receive Complete message50.According to the obtained [dynamic account hash value], the corresponding [dynamic login secret hash value] and [system identification ciphertext] are extracted.51.This time and the previous [dynamic login secret hash value], this time and the previous [system identification ciphertext]52.Request calculation [device–person–OTP self-certification code]53.Calculate [Device–person–OTP self-certificate Code] and return it54.Receive [Device–person–OTP self-certificate Code]55.Require calculation of [system identification restore ciphertext]56.Calculate [system identification restore ciphertext] and return57.Receive [system identification restore ciphertext]58.Transfer [Device–person–OTP Self-Authentication Code], [System Identification Restore Ciphertext]59.Transfer [Device–person–OTP Self-Authentication Code], [System Identification Restore Ciphertext]Accepting OTP and calculating server authentication information by user
60.Receive [Device–person–OTP Self-Authentication Code], [System Identification Restore Ciphertext]61.Display prompt for input [OTP]62.Receive [OTP] newsletters63.Read [OTP]64.Enter [OTP]65.Receive [OTP]66.Transfer [OTP] and request to calculate its secondary hash value67.Receive [OTP] and calculate its secondary hash value, and return68.Is there match between [device–person–OTP self-certification code] and [OTP secondary hash value]?69.NG: check it’s over 3 min70.Y: Go to step 45 to re-enter [OTP]71.N: Go to step 28 to retry [OTP]72.OK: Transfer [system identification restore ciphertext] and require return [system identification plaintext]73.Receive [system identification restore ciphertext] and calculate [system identification plaintext], and return74.Receive [system identification plaintext] and display75.Is the [System Identification Clear Text] correct?76.Not correct: Stop execute77.OK: Click continue excute78.Request to return [dynamic login secret]79.Return [Dynamic Login Secret]80.Receive [Dynamic Login Secret]81.Send [Dynamic Login Secret], [OTP], and [Challenge Message], and request to calculate [Device–person–OTP Login Code]82.Receive information and calculate [Device–user–OTP Login code]83.Receive [Device–user–OTP Login code] and sendVerification of user process on server-side
84.Receive [Device–user–OTP Login code]85.Assign route: judged to verify the login code connection86.Forward [Device–user–OTP Login code]87.Receive [Device–user–OTP Login code]88.Transfer [OTP] and [Dynamic Login Secret Hash Value], and require calculation [Login Code Value]89.Receive the information and calculate the [login code verification value], and return it.90.Receive [login code verification value]91.Does the [login code verification value] match the received [login code]?92.Check it is in blacklisted93.OK: Update [Dynamic Account Hash Value], [Dynamic Login Secret Hash Value], and [System Identification Ciphertext]94.OK: Update [Dynamic Account Hash Value], [Dynamic Login Secret Hash Value], and [System Identification Ciphertext]95.Send [Allow Login] message96.Receive [Allow Login] message97.Forward [Allow Login] messageSynchronization of user- and server-side for updating verification information
98.Receive [Allow Login] message99.Request to update [Dynamic Authentication Secret], [Dynamic Account Self-Authentication Value], and [Dynamic Connection Secret]100.Update [Dynamic Authentication Secret], [Dynamic Account Self-Authentication Value], and [Dynamic Connection Secret]101.Send updated information messages102.Receive news of updated information103.Dispatch routing: just pass the authentication connection104.The notification indicates that the client has been synchronized105.Receive synced messages106.Annotation is synchronized107.Send a note-and-sync message108.Receive a bookmarked sync messageEstablishing secure connection by user and server to transmit information
109.Request to provide services in the SSL environment to the medical system in accordance with [URL]110.Receive service needs111.Create SSL connect112.Provide medical information required113.Receive medical information114.Transfer of medical information115.Open browser and confidential App116.Monitor Browser117.Receive medical information and display it by browser118.Read medical information119.Decide either to end the system or to enter other medical inquires120.The browser receives the user’s decision. If it is the end of the system, go to step 130121.The browser transmits the medical requirements122.Receive medical needs123.Assign route: judged as authenticated connection124.Transfer medical needs125.Receive medical needs126.Provide medical information required127.Receive medical information128.Transfer of medical information129.Go to step 118End of medical HIS system usage by user-side
130.Close Browse131.Clear the data stored in the browser132.Disconnect App133.Close security App

## 5. System Design Analysis and Evaluation

### 5.1. Two-Factor Authentication

One-time password (OTP) is mainly used to perform the two-factor authentication function. Thus, once the user is required to enter one-time password, which secures the login mechanism and protects the user data from being stolen by others. Hence, the password can only be used once. Such mechanism also applies to banking industry. Moreover, we use HOTP, TOTP, and Google Authenticator.
HOTP refers to the HMAC-based one-time password. The algorithm is defined in RFC4226. There are three things that must be known:K: Secret keyC: Random number to ensure that each hash function result is differentHash algorithm: The example we provided above uses SHA1We only need to pair the key with the random information and input in the hash algorithm for further calculation.TOTP is an advanced version of the HOTP algorithm. The main difference is that the time used to generate a different one-time password. As the password will change over time, there is no need to specifically concern about the password retention time limit. Because the password will automatically expire before time is up, the user can save space in the database and server. Only the C value will be given a parameter. This value automatically increases after the time expires, so the same HOTP algorithm automatically changes the parameters of C over time, and produce another passwordGoogle provides Google Authenticators, which only needs to enter the secret key to the Google Authenticator App in order for performing dual authentication function. Users can download the App and scan QR code or enter the secret key (password). TOTP password will be automatically generated, where the users only need to perform verification function

### 5.2. Design Analysis

The purpose of this study is to enhance the security of medical staff accessing medical information outside the hospital through an existing medical information system. Random Oracle Model (ROM) is only a scientific and safety proof of information security mechanism.

In the “Random Oracles are Practical: A Paradigm for Designing Efficient Protocols” model [25] there are several recommendations for designing a suitable P protocol:(1)Plan and provide correctness of P in ROM(2)In the actual scenario, the hash function can be replaced with Oracle

Random Oracle can be regarded as a perfect hash function. It is an abstract concept and cannot be implemented. Therefore, the function h in the actual scenario must be appropriately selected to replace the theoretical Oracle, without causing the gap in theory and actual program. The appropriate selection criterion can be simplified for the use of algorithms that are public and approved for security.

The National Science Council research project “Research on the Safety Certificate based on the Pass-Through Password Authentication Agreement” (plan number: NSC 90-2213-E-006-100-) implemented by the Department of Information Engineering, National Cheng Kung University, reorganized the certification process for the identification mechanism proposed in this study, as described in Figure 23.
(1)Definition: The scope of security is defined by the authentication mechanism. The major focus is to identify the security assumptions. If there is more than one assumption in the mechanism, all must be identified.(2)Model: Stimulating a possible attack based on a defined security range to identify the attacker. The simplest action is to control the network and the execution protocol against the attacker. When there is a higher possibility of attack, it is difficult to ensure that the authentication mechanism is safe.(3)Proof: The falsification method may indicate the nonexistence of the attacker. According to the attack method and situation, the probability method is used to confirm if the attacker can diffuse into the authentication mechanism, it can be based on the attack method used by the attacker.

According to the ROM theory, the proof of the authentication mechanism is shown in Figure 24.
(1)Definition: Define the degree of security and find a formula cited theory.(2)Degree of safety: Stealing network information, encrypting card information, and identifying database information cannot be cracked.(3)Theoretical: AES security, the hash function cannot reverse the input value, and unpredictability of random numbers.(4)Model: Shape the attacker and simulate possible attack methods based on the defined scope.(5)Attacker’s ability: The attacker invades the authentication database, listens to the network, and invades the encryption card.
The attacker selects a random number r and sends it to the user.The user uses the secret Si + 0 as an initial key, XOR’s r and Si + n. Then, h(Si + n) is used to perform AES encryption as a result of XOR operation, and AED encryption on the hash value h(Sj + n) of a newly selected authentication secret Sj+n to obtain Ri+n and Cj + n. It is authenticated by the authenticator, and then Ri+n, Cj+n, while it is verification result Vi + n. Ri + n r and Cj + n h(Sj + n) is the result of nth encryption, and starts from zero, while each time after calculation, Sj+n is used as the next key.Step (2) is repeated by considering that the attacker may get c = {Ri + n, Cj + n, Vi + n}.The attacker selects h(S0) and h(S1) from h(Si + n), hands it to the user along with r, and obtains verification information h(Sk) from the authenticator.The user performs an XOR operation on r and the user authentication secret Sk, and then uses H(Sk) as the key to perform AES encryption as a result of the XOR operation, and also randomly takes one from h(S0) and h(S1), while h(Sb) indicates that AES encryption is performed and response message C={ERh(Sk) (Sk⊕r), ERh(Sk) (h(Sb))} is obtained, which is verified by the authenticator.The message used by the authenticator to verify the change after verification is h(Sb).The user leaks C and its verification result to the attacker, and then uses Sb as the authentication secret used for next login.Can the attacker guess the authentication secret used for the next login (S0 or S1) based on the verification results of step (3), as well as C and C’s verification results?

### 5.3. Performance Evaluation

The performance of the system runtime is critical, including multiple calculations of AES 256 and SHA-256, which are required for system and tablet devices in the authentication process, while security control module is required for the medical system in the TLS/SSL environment.

Regarding the performance of multiple calculations of the cryptographic algorithm, the burden on the tablet device is the Micro SD encryption card. The burden on the system-side is the Micro SD HSM card. It has been verified that its performance is sufficient for instant VoIP encryption and decryption. In the Micro SD performance specification, it is necessary to complete the related operations including Diffie–Hellman key exchange system, RSA encryption and decryption, RSA electronic signature creation, and verification within 3 s. These mechanisms are extravagant in terms of multiple calculations of the modulus index and require high computational cost mechanism in the field of information security technology.

Drawing on the authentication mechanism of this study, the AES 256 and SHA-256 are bit-switched operations, and the computational cost is lower than the calculation of the modulus index. The Micro SD encryption card performs an AES 256 once and can be completed in 0.005 s. The actual time still depends on the length of the encrypted message. The two messages with large differences in message lengths also have significant differences in execution time. At the same time, the performance of the Micro SD HSM card is about five times better than that of the encryption card.

The performance was evaluated at AES 256 execution speed of 0.05 s, and the calculation cost of SHA-256 is considered to be the same as the execution of AES 256. The number of calculations performed on the Micro SD encryption card of the mobile device is less than 20 times. Therefore, the total execution time of the encryption card is 1 s. If I/O processing time is added, it is expected that the calculation required for the authentication procedure can be completed in 1.5 s. The performance of the HSM card is then evaluated. Assuming that the number of people using the medical system at the same time is 200, and 20 of them are identified at the same time, the AES 256 is required to be executed 20 times for each authentication, and 0.01 s for an AES 256. The calculation time required for the HSM card is ~4 s. In order to respond to possible performance concerns, the cryptographic calculation module on the planning system-side is equipped with two HSM cards. In addition to performance considerations, it is also used as a mutual backup.

The above performance analysis data is obtained under strict conditions. The results show that the performance bottleneck should be in the bandwidth of the network and the speed at which the SMS transmits the OTP. The usage efficiency in the TLS/SSL environment is explained further. According to Microsoft’s internal use of IIS’s built-in TLS/SSL, thousands of users can go online at the same time. It is generally believed that the performance of software TLS/SSL should be sufficient to meet the needs of 200 people to access the medical system at the same time. Software TLS/SSL and its performance is related to the host equipment. In addition to adjusting the parameter settings, TLS/SSL can be enabled on the external network to improve the performance of the operation. If the software TLS/SSL cannot be used, there are hardware products on the market, and the system can be added to the TLS/SSL environment for encryption and decryption without affecting the system configuration.

## 6. Prototype Design

This research has built a prototype system using the platform of Hivocal – A Taiwanese company. Hence, we added OTP that uses Micro-SD to generate immediacy in our study. The cost of application increased. However, we believe that it is necessary to do so and guarantee the safety in order to protect the personal data. In this prototype system (see Figure 25), as long as the Micro-SD card in the Mobile phone screen is moved to the red slot on the left side, the window will pop up to enter the password. When a user enters the password and presses “OK” button, the operation is performed. If the password is incorrect, “Authentication is incorrect, please re-enter” message will appear, and notify the user through system or webpage.

## 7. Conclusions and Suggestion

In the face of the information security needs of mobile devices using HIS systems outside the hospital, the existing security protection programs, or the use of public key systems, are not efficient, or require high costs. There is an immediate risk that the mobile device is lost or the identification database is compromised. The information security nature of the security system architecture and mechanisms outlined in this study include the following.
The system-side must adopt an architecture that can protect the HIS systemUsers and their devices must provide HIS information access services through authenticationThe authentication mechanism must be able to distinguish whether to attempt malicious loginIt is necessary to avoid the user inputting the website and reducing the chance of success of the phishing websiteThe system-side is used to verify the authentication information of the users and their devices.The data transmitted on the network must be encrypted to prevent being recordedIt is necessary to prevent the medical information temporarily stored in the mobile device from being stolenA malicious third-party who steals a mobile device must be prevented to impersonate the user’s login system from the information stored in the deviceVulnerability of the user’s passphrase must be reinforced, which is easily guessed or recorded by a Trojan horse programSecurity measures must be implemented with efficiency and cost considerations

Mobile devices are extensively used for diverse operations in different industries, including manufacturing, accounting, and so on. To get real-time information regarding decision-making, secure mobile device connectivity is a must. Therefore, this research is not only considered for the application of hospital industry but can also be used in other service and manufacturing industries.

In summary, good technical planning can fully assist management measures and effectively prevent the personal information leakage incidents. Through the discussion of this preliminary research, more complete structure of the out-of-hospital HIS data access system is proposed from the management and security perspectives. Although Taiwan ranks among the best in the global medical industry, it can provide more intimate and timely medical services for medical institutions, which can ultimately reduce the security risks for medical practitioners and medical institutions.

## Figures and Tables

**Figure 1 sensors-19-02628-f001:**
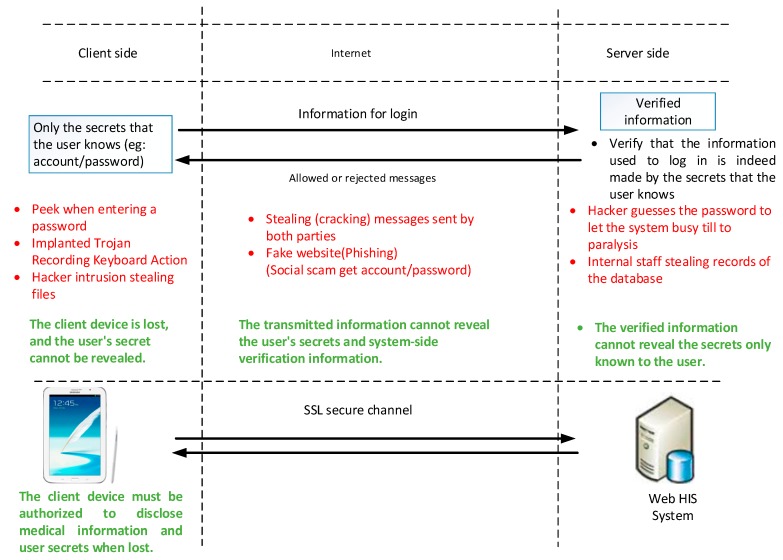
Diagram of mobile device usage outside the hospital.

**Figure 2 sensors-19-02628-f002:**
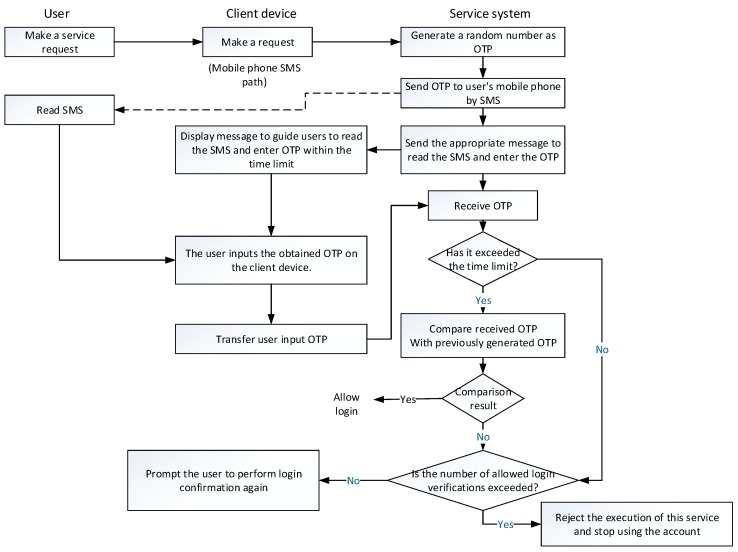
Short message service (SMS) OTP user authentication mechanism.

**Figure 3 sensors-19-02628-f003:**
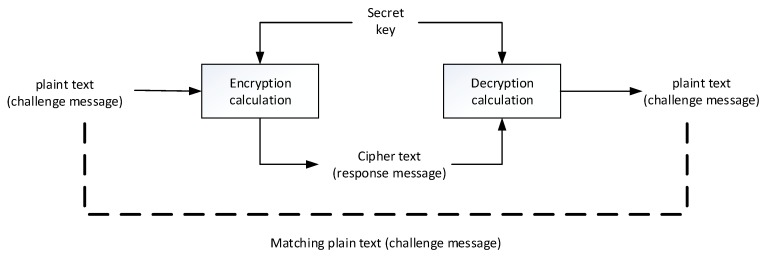
Challenge and response concept.

**Figure 4 sensors-19-02628-f004:**
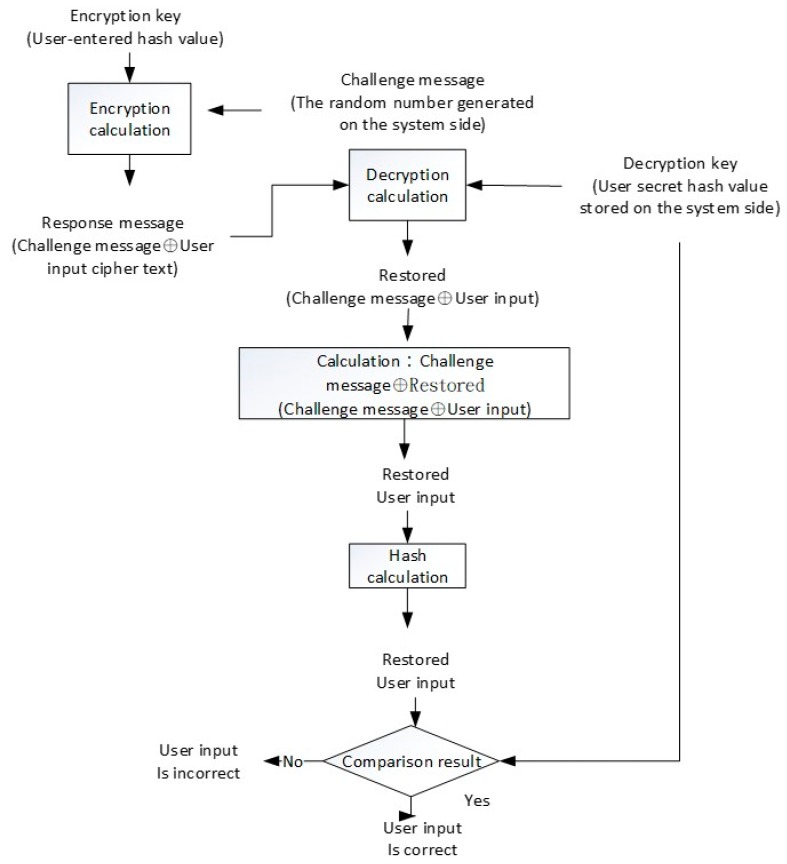
Concept of security challenges and response.

**Figure 5 sensors-19-02628-f005:**
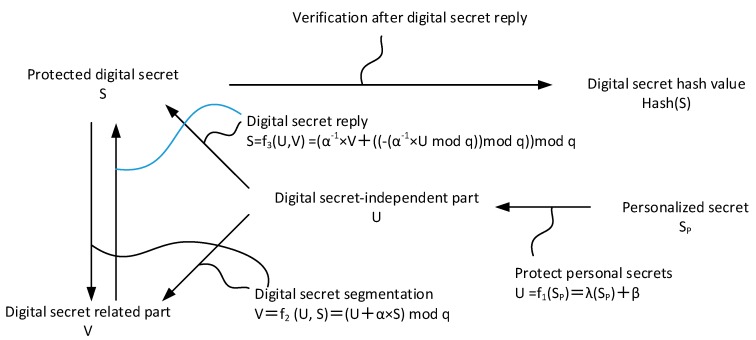
Schematic diagram of the technical idea for digital secrets protection.

**Figure 6 sensors-19-02628-f006:**
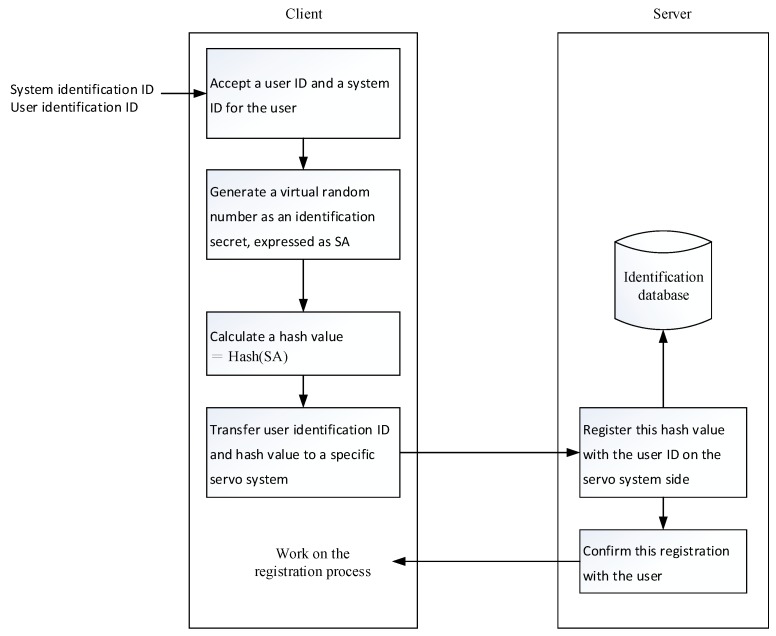
“By identifying users who use randomly generated authentication secrets and personalized secrets”. Registration process one.

**Figure 7 sensors-19-02628-f007:**
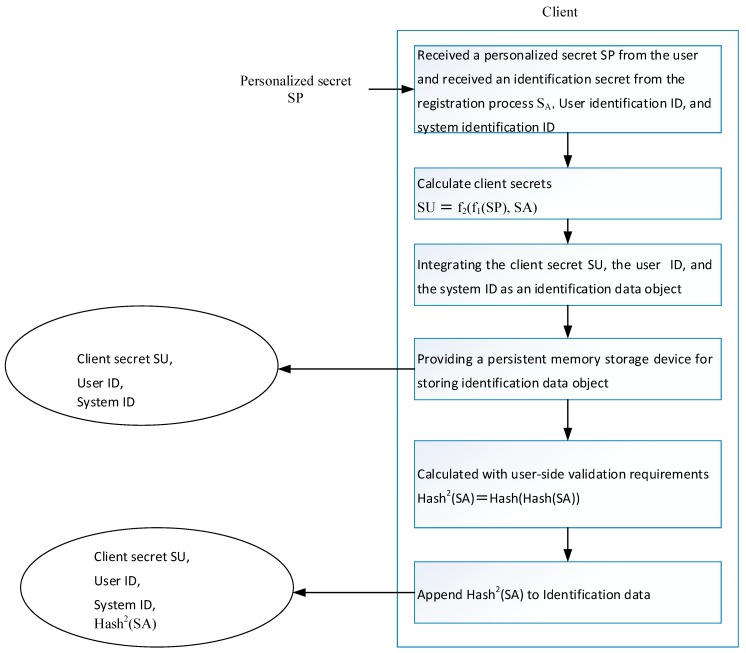
“By identifying users who use randomly generated authentication secrets and personalized secrets”. Registration process work 2.

**Figure 8 sensors-19-02628-f008:**
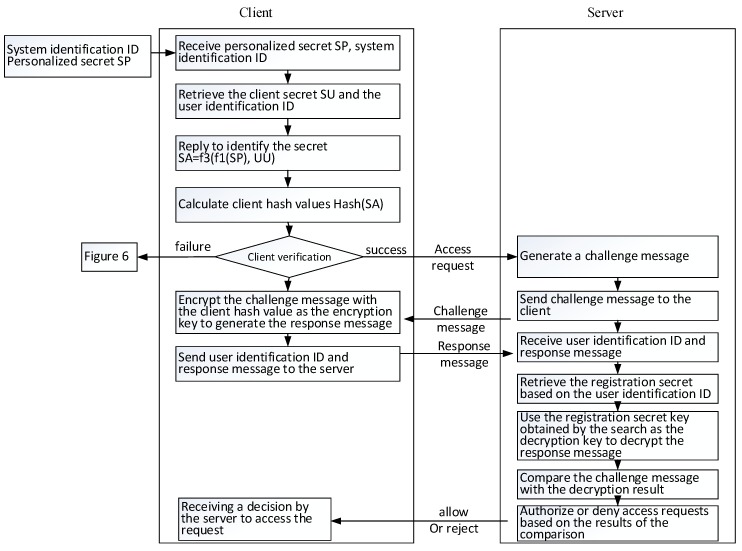
User authentication by linking randomly generated authentication secrets and personalized secrets, login program with user authentication.

**Figure 9 sensors-19-02628-f009:**
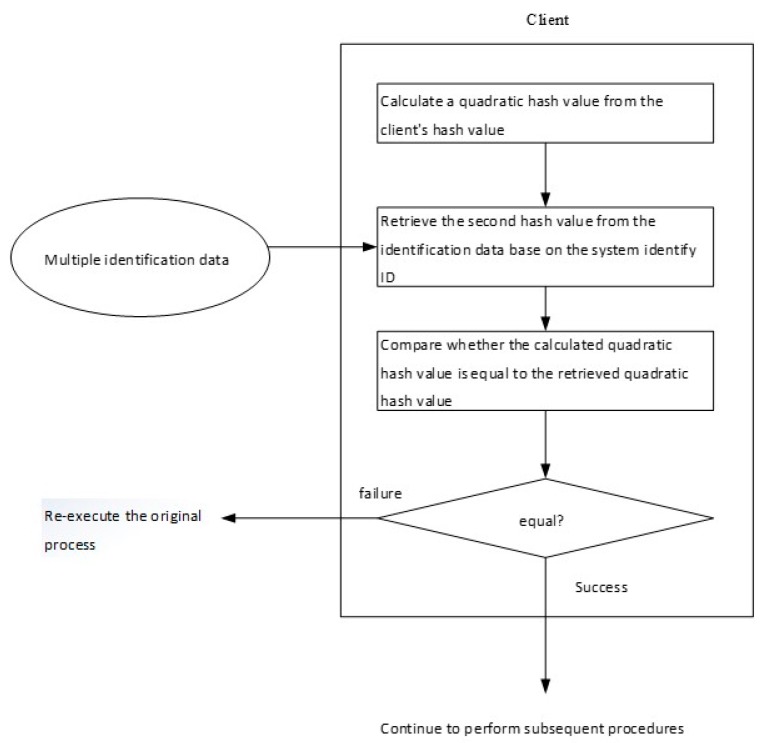
User-side verifier authenticating users by randomly connecting authentication secrets with personalized secrets.

**Figure 10 sensors-19-02628-f010:**
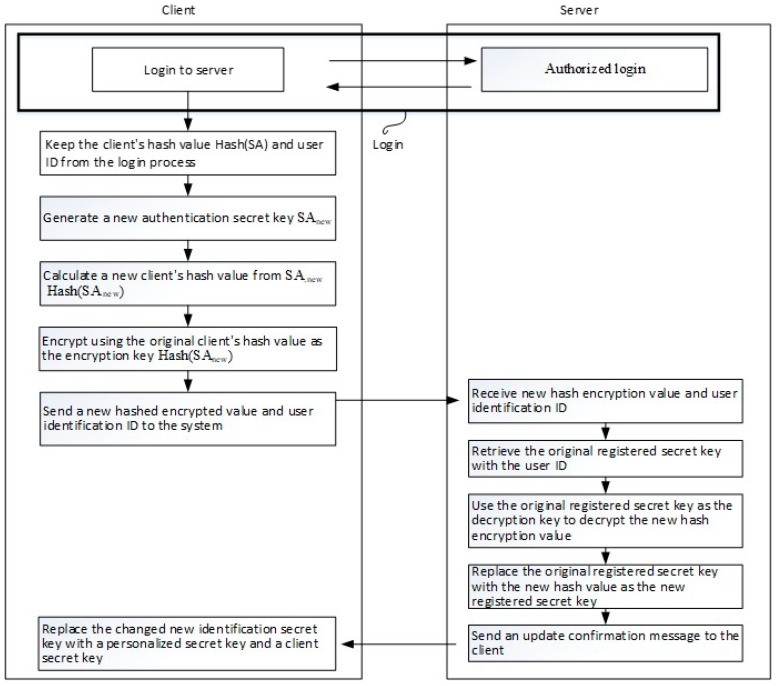
Procedures for authenticating users who authenticate randomly generated secrets and personalized secrets.

**Figure 11 sensors-19-02628-f011:**
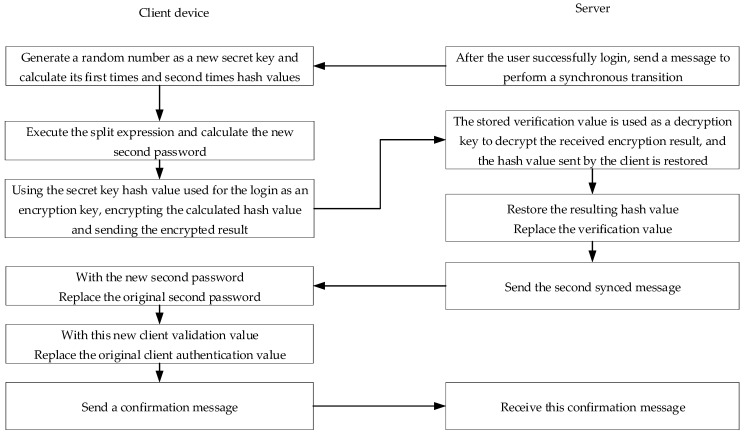
Synchronous shift process.

**Figure 12 sensors-19-02628-f012:**
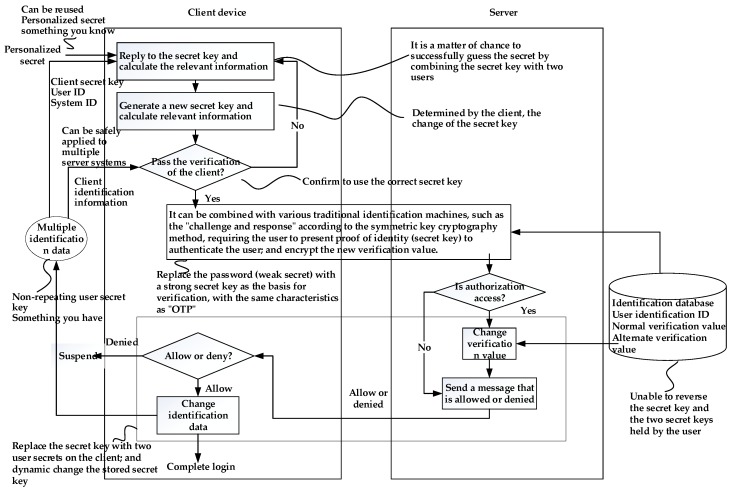
“Two-factor authentication using repeatable passphrases and nonrepeating user secrets”. Characteristic.

**Figure 13 sensors-19-02628-f013:**
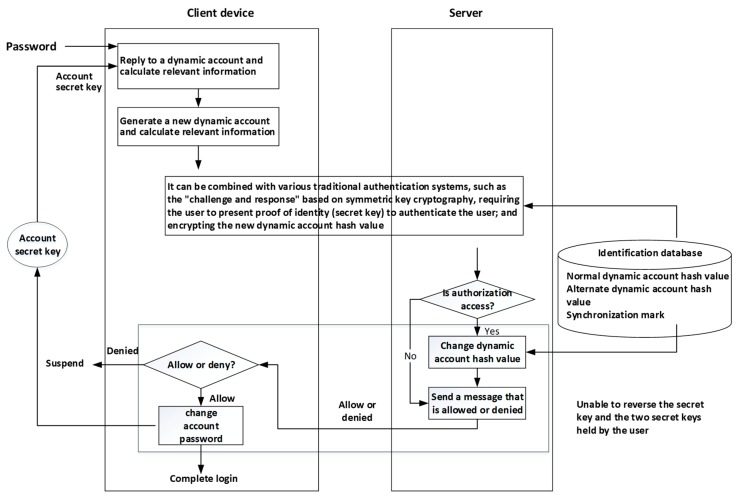
User identification technology and system characteristics of dynamic account.

**Figure 14 sensors-19-02628-f014:**
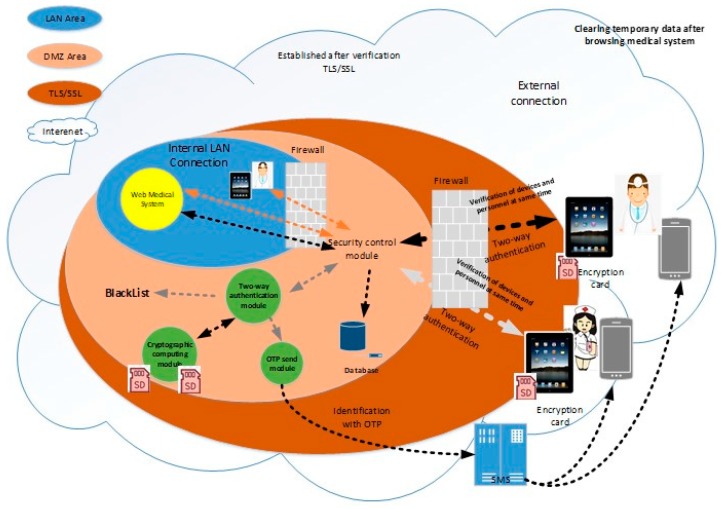
System architecture diagram.

**Figure 15 sensors-19-02628-f015:**
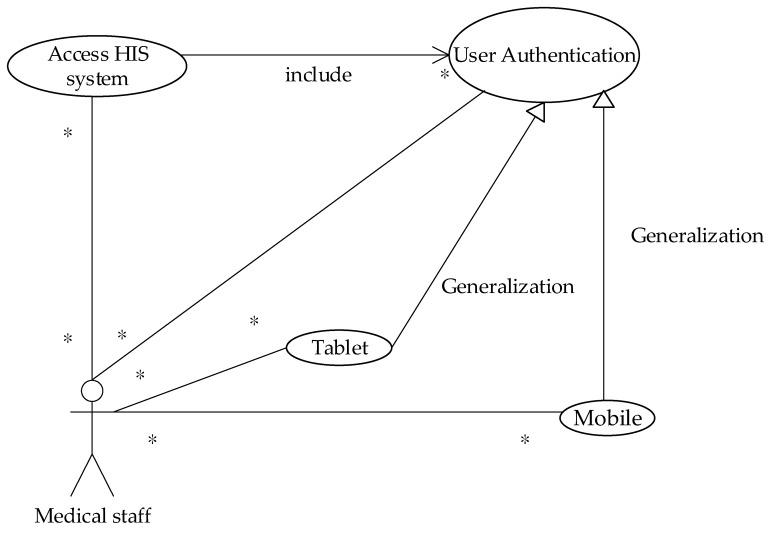
Use case diagram of interaction between medical staff and HIS system.

**Figure 16 sensors-19-02628-f016:**
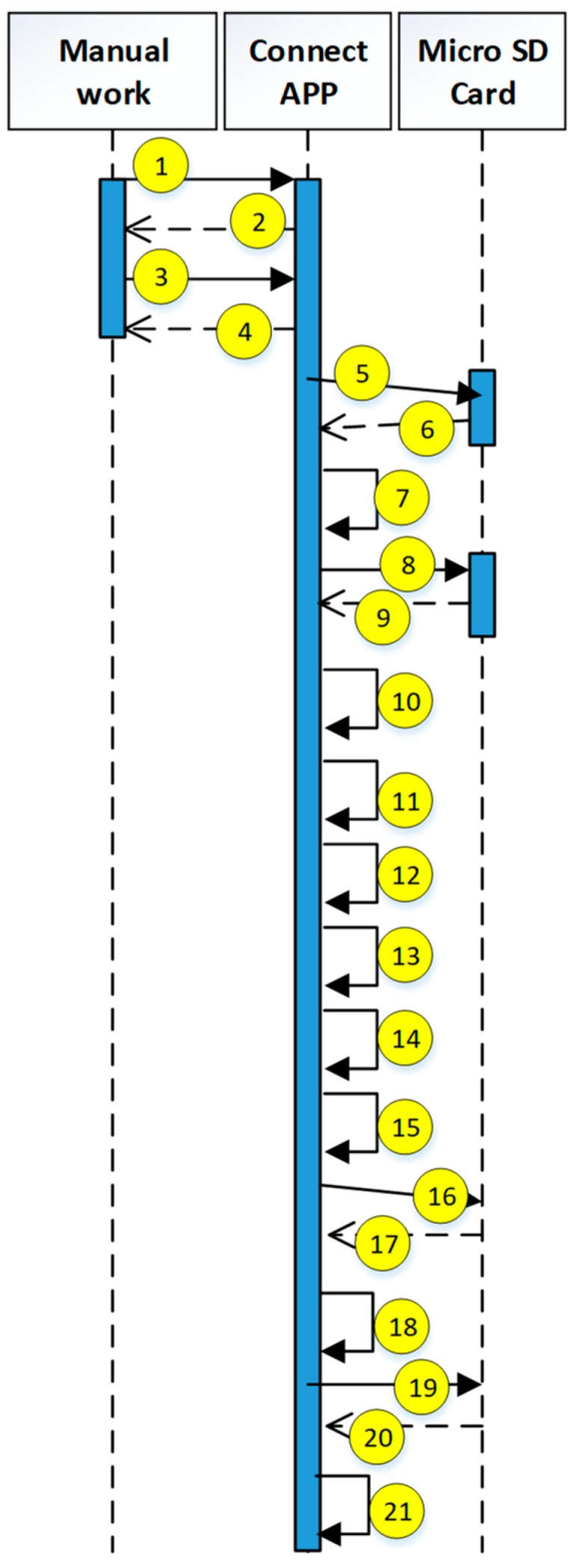
User-to-server connection request diagram 1.

**Figure 17 sensors-19-02628-f017:**
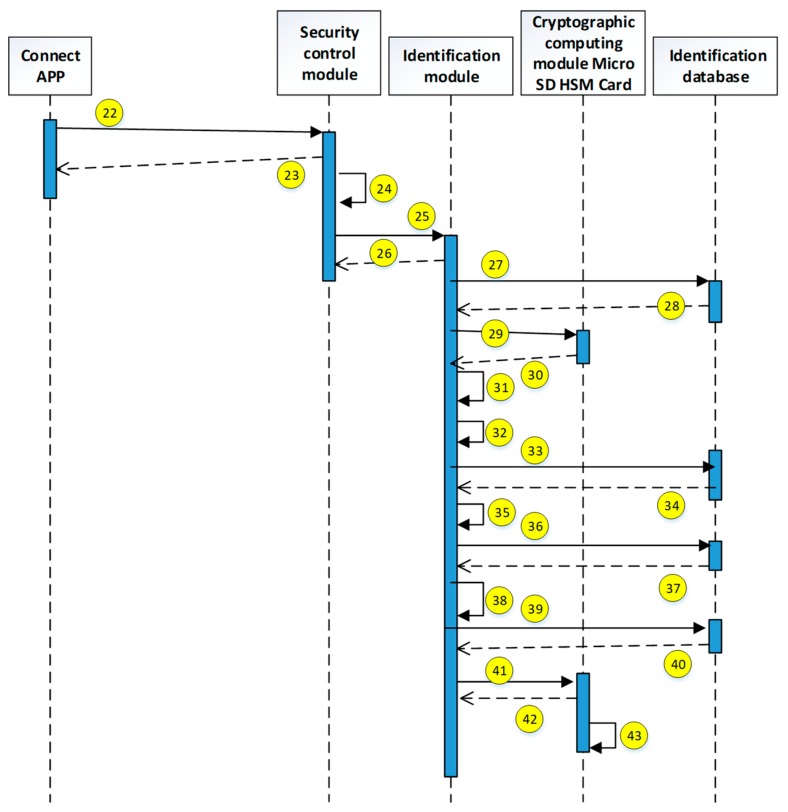
OTP generation by the server after nonblacklist verification diagram 2.

**Figure 18 sensors-19-02628-f018:**
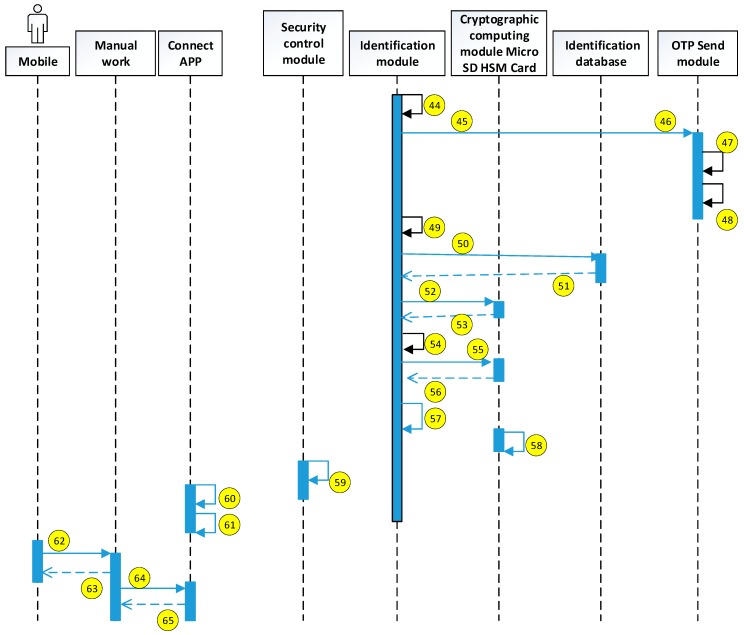
System operation sequence diagram 3.

**Figure 19 sensors-19-02628-f019:**
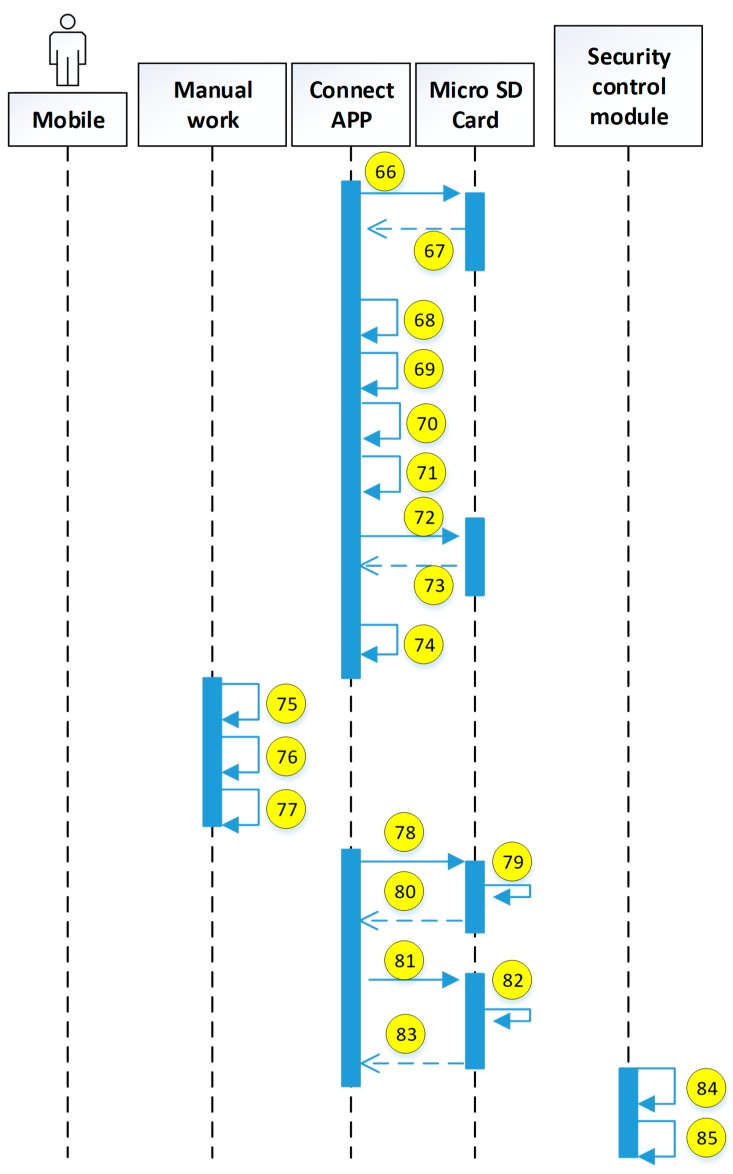
System operation sequence diagram 4.

**Figure 20 sensors-19-02628-f020:**
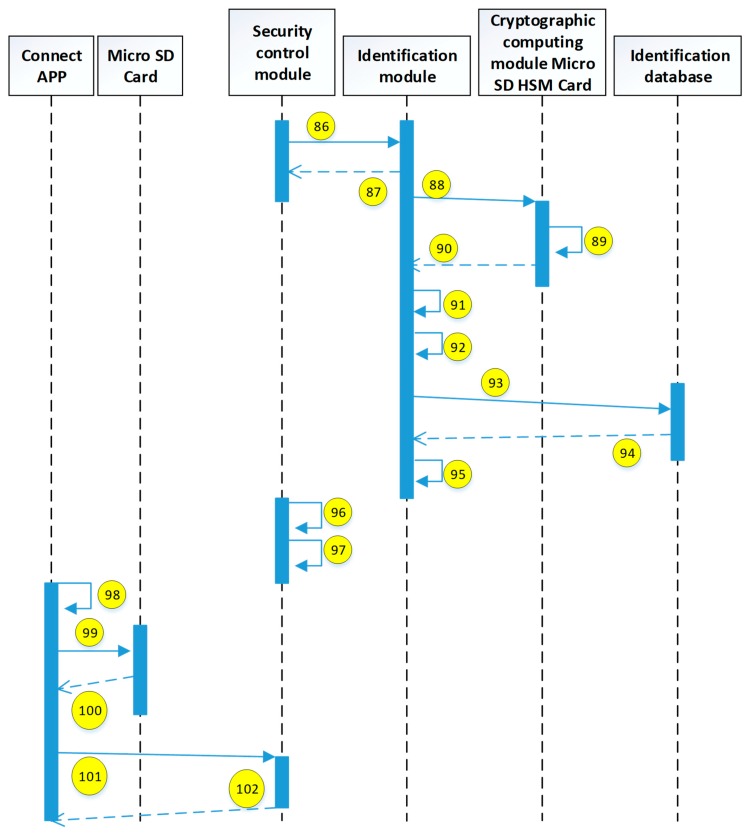
System operation sequence diagram 5.

**Figure 21 sensors-19-02628-f021:**
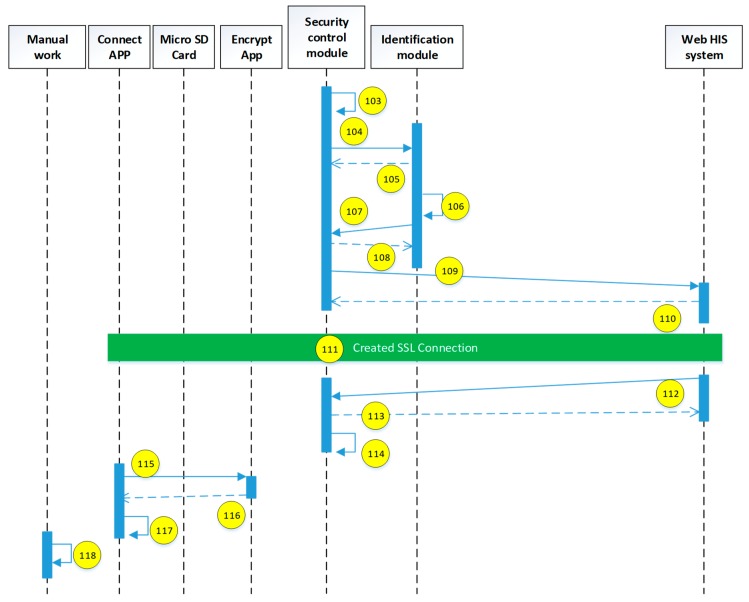
System operation sequence diagram 6.

**Figure 22 sensors-19-02628-f022:**
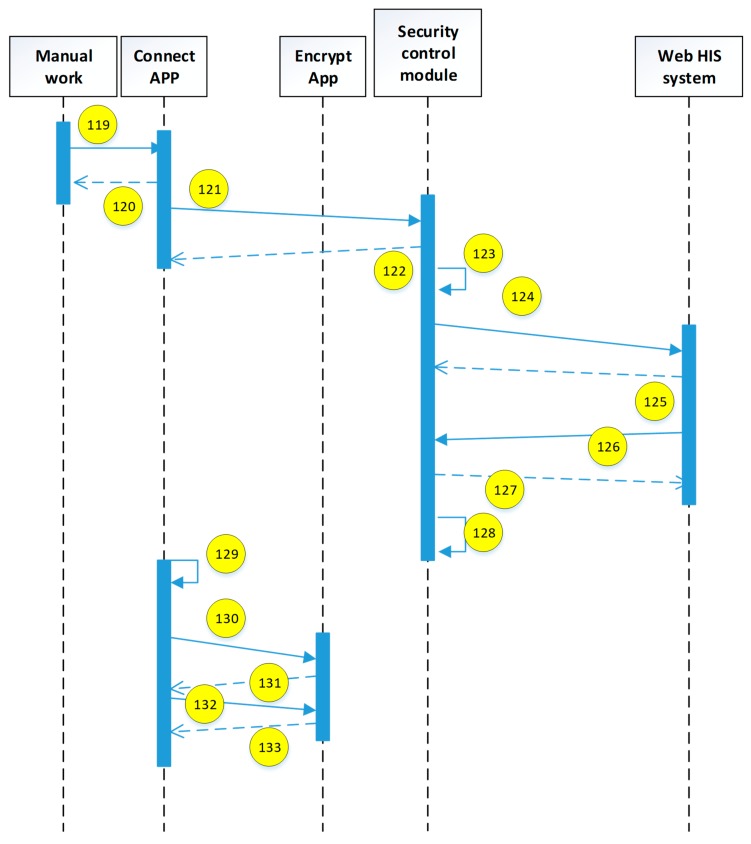
System operation sequence diagram 7.

**Figure 23 sensors-19-02628-f023:**
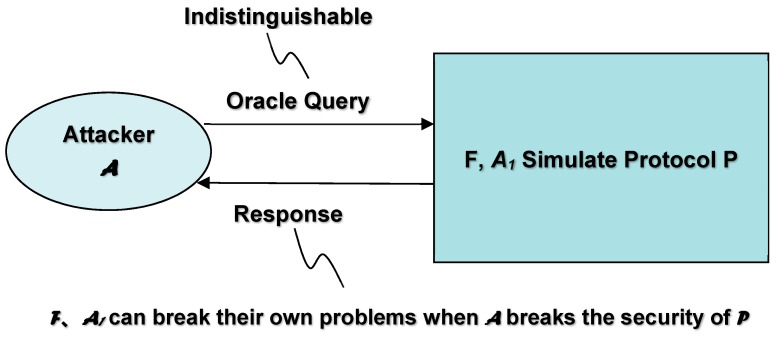
Concept of ROM proof model. (Soure: Base on Prof. Hwang, National Science Council Research Project report “Study on Safety Certificates Based on Pass-Through Password Authentication Agreement”, plan no: NSC 90-2213-E-006-100-).

**Figure 24 sensors-19-02628-f024:**
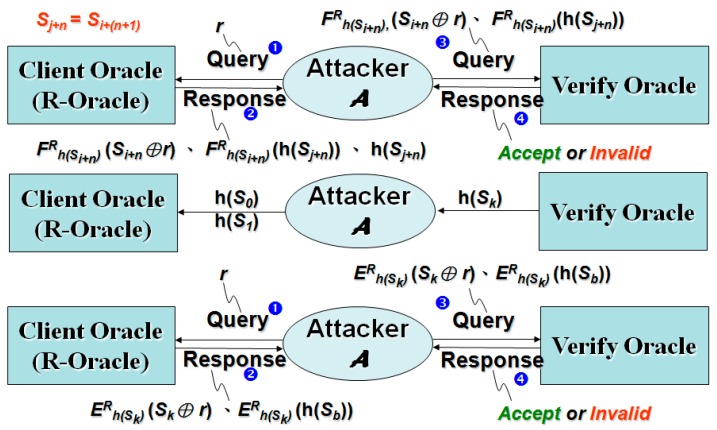
ROM proof model of authentication mechanism.

**Figure 25 sensors-19-02628-f025:**
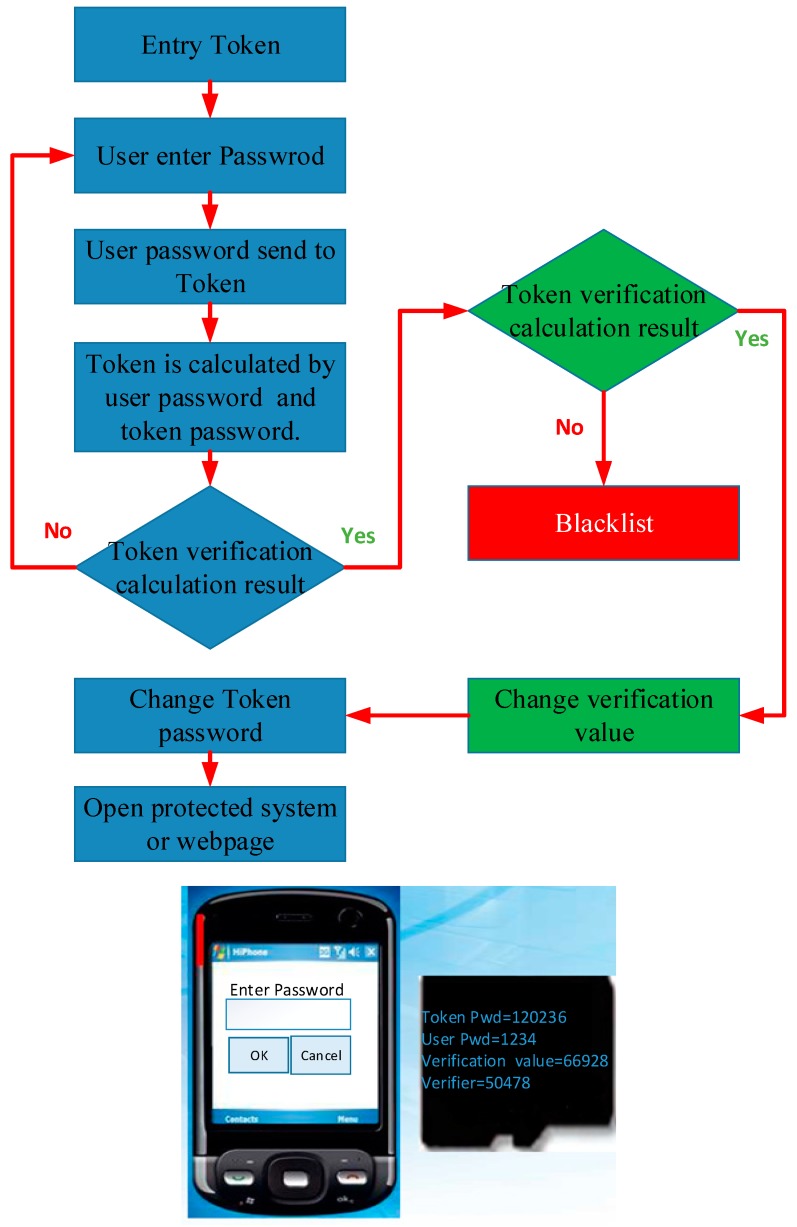
The flowchart and the interface of the user-side prototype.

**Table 1 sensors-19-02628-t001:** Module description.

Category	Sub-Module	Module Content
Server-side module	Security control module	Differentiating between the inside and outside of the hospital Medical system agent Connection routing controll
	Two-way authentication module	Simultaneous identification of users, tablets and cards Verification data change handler Connection accidental interruption of the alternate authentication procedure System identification information protection and recovery
	OTP send module	AES 256 SHA-256 Call SDencrypter-HSM Encryption card
	Cryptographic computing module	Generate random numbers Using random numbers as the basis for 6-code OTP calculation Send OTP
	Management module	User management Out-of-hospital connection register apply and review Stop, reuse, and logout of connection permission Encryption card distribution record Blacklist add, remove and query
User-side Module	Connection App	In-of-hospital domain connection Out-of-hospital domain connection Call encryption card Dynamic account calculation User, tablet, encryption card self-verification Login data change processing Server-side verify Open browser Call to apply to enable the function of the out-of-hospital connection
	Encryption App	Clearing temporary data when browser is closed Clear browser temporary data before the mobile device is turned off
	Cryptographic module integration operation	Integration with the PKCS#11 cryptographic module in the encryption card

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
