# Peer review of "Study of Out-Of-Hospital Access to HIS System: A Security Perspective"

_sensors, 2019, doi:10.3390/s19112628_

Reviewer 1 Report

My detailed comments are as follows:

1)      The authors should highlight the contributions of this manuscript in the Introduction Part.

2)      The grammar and the spelling should be carefully checked.

3)      The authors make a valiant effort to comply with all the usual publication norms but the format should be improved by using a professional layout tool.

4)      There are some unclear figures, Figure1, Figure 8, Figure 12, and Figure 13 etc.

5)      The system methods are sound. However, the paper can be improved by comparing with more recent works on this topic in Part 5 System design analysis and evaluation.

6)      The authors should add security proof or analysis of the proposed system.

7)      More discussion about their future works should be given in conclusion.

8)      Some new references of this topic should be considered.

[1]Fine-grained multi-authority access control in IoT-enabled mHealth, Annals of Telecommunications, 2019

[2] Enhancing privacy and availability for data clustering in intelligent electrical service of IoT, IEEE Internet of Things Journal, 2018

[3] A Feature based Learning System for Internet of Things Applications. IEEE Internet of Things Journal

Author Response

1)      The authors should highlight the contributions of this manuscript in the Introduction Part.

Answer: The contribution of the research has been added on Page 2, Lines 84~89.

2)      The grammar and spelling should be carefully checked.

Answer: We have edited the grammar mistakes and spell-checks.

3)      The authors make a valiant effort to comply with all the usual publication norms but the format should be improved by using a professional layout tool.

Answer: Re-formatting has been done.

4)      There are some unclear figures, Figure1, Figure 8, Figure 12, and Figure 13 etc.

Answer: Figures 1, 8, 12, and 13 have been modified.

5)      The system methods are sound. However, the paper can be improved by comparing with more recent works on this topic in Part 5 System design analysis and evaluation.

Answer: Section 5 has been divided into 2 sub-sections and two-factor authentication has been explained on Page 25, Lines 830~853.

6)      The authors should add security proof or analysis of the proposed system.

Answer: Security proof analysis has been provided on Pages 26~28, Lines 898~970.

7)      More discussion about their future works should be given in conclusion.

Answer: More discussion has been added on Page 30, Lines 1006~1010.

8)      Some new references for this topic should be considered.

[1]Fine-grained multi-authority access control in IoT-enabled mHealth, Annals of Telecommunications, 2019

Answer: Li et.al [9] has been added to Page 2, Lines 53~56.

[2] Enhancing privacy and availability for data clustering in intelligent electrical service of IoT, IEEE Internet of Things Journal, 2018

Answer: Xiong et al. [10] has been added to Page 2, Lines 56~58.

[3] A Feature based Learning System for Internet of Things Applications. IEEE Internet of Things Journal

Answer: We think that reviewer’s suggestion of the third reference is not suitable for use in this article. This paper is Offloading data classification and anomaly event detection tasks to sink nodes in sensor networks can reduce the computing complexity, lower remote communication loads, and improve the response time for the delay-sensitive IoT applications. If the encryption and decryption operation is performed, then his load must increase, so we have determined that this reference file is not suitable for this article.

Reviewer 2 Report

The authors reviewed the security threats that can appear in a case when the Hospital Information System is accessed from mobile devices located outside the hospital. The authors also analyzed the security mechanisms that can be used to minimize threats. Also, the architecture of a hospital information system using security mechanisms is proposed.

The main problem with this paper is that it is purely theoretical. The real system (or even its prototype) is nonexistent (or at least it is not presented in the paper), so there is no empirical evidence that the proposed solution is valid and what are its possible weak points. The authors have not provided any empirical evidence that their solution is better than the existing ones.

My suggestion is to implement the prototype of the system, perform some experiments showing the pros and cons of the proposed architecture and security mechanisms.

Author Response

The authors reviewed the security threats that can appear in a case when the Hospital Information System is accessed from mobile devices located outside the hospital. The authors also analyzed the security mechanisms that can be used to minimize threats. Also, the architecture of a hospital information system using security mechanisms is proposed.

The main problem with this paper is that it is purely theoretical. The real system (or even its prototype) is nonexistent (or at least it is not presented in the paper), so there is no empirical evidence that the proposed solution is valid and what are its possible weak points. The authors have not provided any empirical evidence that their solution is better than the existing ones.

My suggestion is to implement the prototype of the system, perform some experiments showing the pros and cons of the proposed architecture and security mechanisms.

Answer: The prototype design and description has been added as Section 6, on Pages 28 and 29, Lines 971~981 (including Figure 17).

Round  2

Reviewer 2 Report

The authors added a description of a simple prototype (it is not clear which algorithms are implemented and which are not – the authors showed only a screenshot with password request). However, there are still no results of experiments verifying whether the prototype works at all and whether it is secure in the selected scenarios. Please perform extensive experiments, analyze the results, and on that basis, analyze the pros and cons of the proposed approach. Also, please analyze experimentally the computation time needed to run the selected security algorithms.

Also, the English language should be vastly improved - there are lots of grammar, style, and interpunction errors.

Author Response

1. The authors added a description of a simple prototype (it is not clear which algorithms are implemented and which are not – the authors showed only a screenshot with password request). However, there are still no results of experiments verifying whether the prototype works at all and whether it is secure in the selected scenarios. Please perform extensive experiments, analyze the results, and on that basis, analyze the pros and cons of the proposed approach. Also, please analyze experimentally the computation time needed to run the selected security algorithms.

Answer: Section 4.5 Steps of the operating system on Pages 24~34 and Table 2 describe the prototype in more details. Table 1 describes the modules in the system architecture, on Pages 21~22.

Reference [25] has been added to give extra details about the theoretical proof of security. 

2. Also, the English language should be vastly improved - there are lots of grammar, style, and interpunction errors.

Answer: We have eliminated all potential language issues.

Round  3

Reviewer 2 Report

The new Section 4.5 should be rewritten - please use the UML interaction diagrams (the Sequence Diagram, the Collaboration  Diagram) and maybe also other UML diagrams like the Class Diagram, Component Diagram, and Use Case Diagram. The description in the form of a numbered list of actions and tables is unreadable for the reader, and it should be changed. If You do not have a fully functional prototype, please at least describe the idea of the system using state-of-the-art software engineering tolls like UML.

Is it really "operating system" that is described in section 4.5? The term "operating system" is used for systems like Linux, Widows, etc. Do You plan to design the whole new operating system?

Author Response

The new Section 4.5 should be rewritten - please use the UML interaction diagrams (the Sequence Diagram, the Collaboration Diagram) and maybe also other UML diagrams like the Class Diagram, Component Diagram, and Use Case Diagram. The description in the form of a numbered list of actions and tables is unreadable for the reader, and it should be changed. If you do not have a fully functional prototype, please at least describe the idea of the system using state-of-the-art software engineering tolls like UML.

Is it really "operating system" that is described in section 4.5? The term "operating system" is used for systems like Linux, Windows, etc. Do you plan to design the whole new operating system?

Answer: Thanks so much to the reviewer for giving very constructive comments, which led our manuscript to become better. Please note that the operating steps of the prototype system have been explained by diagrams on Figures 15~21. In addition, “operating system” in the heading of Section 4.5 has been modified and the whole title has been changed on Page 24, Line 820.
